# SOAR: SECOND-ORDER ADVERSARIAL REGULARIZATION

## ABSTRACT

Adversarial training is a common approach to improving the robustness of deep neural networks against adversarial examples. In this work, we propose a novel regularization approach as an alternative. To derive the regularizer, we formulate the adversarial robustness problem under the robust optimization framework and approximate the loss function using a second-order Taylor series expansion. Our proposed *second-order adversarial regularizer* (SOAR) is an upper bound based on the Taylor approximation of the inner-max in the robust optimization objective. We empirically show that the proposed method improves the robustness of networks against the $\ell_\infty$ and $\ell_2$ bounded perturbations on CIFAR-10 and SVHN.

## 1 INTRODUCTION

Adversarial training (Szegedy et al., 2013) is the standard approach for improving the robustness of deep neural networks (DNN), or any other model, against adversarial examples. It is a data augmentation method that adds adversarial examples to the training set and updates the network with newly added data points. Intuitively, this procedure encourages the DNN not to make the same mistakes against an adversary. By adding sufficiently enough adversarial examples, the network gradually becomes robust to the attack it was trained on. One of the challenges with such a data augmentation approach is the tremendous amount of additional data required for learning a robust model. Schmidt et al. (2018) show that under a Gaussian data model, the sample complexity of robust generalization is $\sqrt{d}$ times larger than that of standard generalization. They further suggest that current datasets (e.g., CIFAR-10) may not be large enough to attain higher adversarial accuracy. A data augmentation procedure, however, is an indirect way to improve the robustness of a DNN. Our proposed alternative is to define a regularizer that penalizes DNN parameters prone to attacks. Minimizing the regularized loss function leads to estimators robust to adversarial examples.

Adversarial training and our proposal can both be formulated in terms of robust optimization framework for adversarial robustness (Ben-Tal et al., 2009; Madry et al., 2018; Wong & Kolter, 2018; Shaham et al., 2018; Sinha et al., 2018). In this formulation, one is seeking to improve the *worst-case* performance of the model, where the performance is measured by a particular loss function $\ell$. Adversarial training can be understood as approximating such a worst-case loss by finding the corresponding worst-case data point, i.e., $x + \delta$ with some specific attack techniques. Our proposed method is more direct. It is based on approximating the loss function $\ell(x + \delta)$ using its second-order Taylor series expansion, i.e.,

$$\ell(x + \delta) \approx \ell(x) + \nabla_x \ell(x)^\top \delta + \frac{1}{2} \delta^\top \nabla_x^2 \ell(x) \delta,$$

and then upper bounding the worst-case loss using the expansion terms. By considering both gradient and Hessian of the loss function with respect to (w.r.t.) the input, we can provide a more accurate approximation to the worst-case loss. In our derivations, we consider both $\ell_2$ and $\ell_\infty$ attacks. In our derivations, the second-order expansion incorporates both the gradient and Hessian of the loss function with respect to (w.r.t.) the input. We call the method *Second-Order Adversarial Regularizer (SOAR)* (not to be confused with the Soar cognitive architecture Laird 2012). In the course of development of SOAR, we make the following contributions:

- We show that an over-parameterized linear regression model can be severely affected by an adversary, even though its population loss is zero. We robustify it with a regularizer that

**exactly** mimics the adversarial training. This suggests that regularization can be used instead of adversarial training (Section 2).

- Inspired by such a possibility, we develop a regularizer which upper bounds the **worst-case** effect of an adversary under an approximation of the loss. In particular, we derive SOAR, which approximates the inner maximization of the robust optimization formulation based on the second-order Taylor series expansion of the loss function (Section 4).

- We study SOAR in the logistic regression setting and reveal challenges with regularization using Hessian w.r.t. the input. We develop a simple initialization method to circumvent the issue (Section 4.1).

- We empirically show that SOAR significantly improves the adversarial robustness of the network against $\ell_\infty$ attacks and $\ell_2$ attacks on CIFAR-10 and SVHN. Specifically, we evaluate using a PGD1000 white-box attack (Madry et al., 2018), transferred PGD1000 attacks, AutoAttack (Croce & Hein, 2020), and SimBA (Guo et al., 2019).

## 2 LINEAR REGRESSION WITH AN OVER-PARAMETRIZED MODEL

This section shows that for over-parameterized linear models, gradient descent (GD) finds a solution that has zero population loss, but is prone to attacks. It also shows that one can avoid this problem with defining an appropriate regularizer. Hence, we do not need adversarial training to robustify such a model. This simple illustration motivates the development of our method in next sections. We only briefly report the main results here, and defer the derivations to Appendix A .

Consider a linear model $f_w(x) = \langle w, x \rangle$ with $x, w \in \mathbb{R}^d$. Suppose that $w^* = (1, 0, 0, \ldots, 0)^\top$ and the distribution of $x \sim p$ is such that it is confined on a 1-dimensional subspace $\{(x_1, 0, 0, \ldots, 0) : x_1 \in \mathbb{R}\}$. This setup can be thought of as using an over-parameterized model that has many irrelevant dimensions with data that is only covering the relevant dimension of the input space. This is a simplified model of the situation when the data manifold has a dimension lower than the input space. We consider the squared error pointwise loss $l(x; w) = \frac{1}{2} |\langle x, w \rangle - \langle x, w^* \rangle|^2$. Denote the residual by $r(x; w) = \langle x, w - w^* \rangle$, and the population loss by $\mathcal{L}(w) = \mathbb{E}[l(X; w)]$.

Suppose that we initialize the weights as $w(0) = W \sim N(0, \sigma^2 \mathbf{I}_{d \times d})$, and use GD on the population loss, i.e., $w(t+1) \leftarrow w(t) - \beta \nabla_w \mathcal{L}(w)$. It is easy to see that the partial derivatives w.r.t. $w_{2,\ldots,d}$ are all zero, i.e., no weight adaptation happens. With a proper choice of learning rate $\beta$, we get that the asymptotic solution is $\bar{w} \triangleq \lim_{r \to \infty} w(t) = (w_1^*, w_2(0), w_3(0), \ldots, w_d(0))^\top$. That is, the initial random weights on dimensions $2, \ldots, d$ do not change.

We make two observations. The first is that $\mathcal{L}(\bar{w}) = 0$, i.e., the population loss is zero. So from the perspective of training under the original loss, we are finding the optimal solution. The second observation is that this model is vulnerable to adversarial examples. An FGSM-like attack that perturbs $x$ by $\Delta x = (0, \Delta x_2, \Delta x_3, \ldots, \Delta x_d)^\top$ with $\Delta x_i = \varepsilon \operatorname{sign}(w_i(0))$ (for $i = 2, \ldots, d$) has the population loss of $\mathbb{E}_{X,W}[l(X + \Delta x); \bar{w})] \approx O(\varepsilon^2 d^2 \sigma^2)$ under the adversary at the asymptotic solution $\bar{w}$. When the dimension is large, this loss is quite significant. The culprit is obviously that GD is not forcing the initial weights to go to zero when there is no data from irrelevant and unused dimensions. This simple problem illustrates how the optimizer and an over-parameterized model might interact and lead to a solution that is prone to attacks.

An effective solution is to regularize the loss such that the weights of irrelevant dimensions to go to zero. Generic regularizers such as ridge and Lasso regression lead to a biased estimate of $w_1^*$, and thus, one is motivated to define a regularizer that is specially-designed for improving adversarial robustness. Bishop (1995) showed the close connection between training with random perturbation and Tikhonov Regularization. Inspired by this idea, we develop a regularizer that mimics the adversary itself. For this FGSM-like adversary, the population loss at the perturbed point is

$$\mathcal{L}_{\text{robustified}}(w) \triangleq \mathbb{E}[l(X + \Delta x; w)] = \mathcal{L}(w) + \varepsilon \mathbb{E}[r(X; w)] \|w_{2:d}\|_1 + \frac{\varepsilon^2}{2} \|w_{2:d}\|_1^2. \quad (1)$$

Minimizing $\mathcal{L}_{\text{robustified}}(w)$ is equivalent to minimizing the model at the point $x' = x + \Delta x$. The regularizer $\varepsilon \mathbb{E}[r(X; w)] \|w_{2:d}\|_1 + \frac{\varepsilon^2}{2} \|w_{2:d}\|_1^2$ incorporates the effect of adversary in exact form.

Nonetheless, there are two limitations of this approach. The first is that it is designed for a particular choice of attack, an FGSM-like one. We would like a regularizer that is robust to a larger class

of attacks. The second is that this regularizer is designed for a linear model and the squared error loss. How can we design a regularizer for more complicated models, such as DNNs? We address these questions by formulating the problem of adversarial robustness within the robust optimization framework (Section 3), and propose an approach to approximately solve it (Section 4).

## 3 ROBUST OPTIMIZATION FORMULATION

Designing an adversarial robust estimator can be formulated as a robust optimization problem (Huang et al., 2015; Madry et al., 2018; Wong & Kolter, 2018; Shaham et al., 2018). To describe it, let us introduce our notations first. Consider an input space $\mathcal{X} \subset \mathbb{R}^d$, an output space $\mathcal{Y}$, and a parameter (or hypothesis) space $\mathcal{W}$, parameterizing a model $f : \mathcal{X} \times \mathcal{W} \to \mathcal{Y}$. In the supervised learning scenario, we are given a data distribution $\mathcal{D}$ over pairs of examples $\{(X_i, Y_i)\}_{i=1}^n$. Given the prediction of $f(x; w)$ and a target value $y$, the pointwise loss function of the model is denoted by $\ell(x, y; w) \triangleq \ell(f(x; w), y)$. Given the distribution of data, one can define the population loss as $\mathcal{L}(w) = \mathbb{E}[\ell(X, Y; w)]$. The goal of the standard supervised learning problem is to find a $w \in \mathcal{W}$ that minimizes the population loss. A generic approach to do this is through empirical risk minimization (ERM). Explicit or implicit regularization is often used to control the complexity of the hypothesis to avoid over- or under-fitting (Hastie et al., 2009).

As shown in the previous section, it is possible to find a parameter $w$ that minimizes the loss through ERM, but leads to a model that is vulnerable to adversarial examples. To incorporate the robustness notion in the model, it requires defenders to reconsider the training objective. It is also important to formalize and constrain the power of the adversary, so we understand the strength of the attack to which the model is resistant. This can be specified by limiting that the adversary can only modify any input $x$ to $x + \delta$ with $\delta \in \Delta \subset \mathcal{X}$. Commonly used constraints are $\varepsilon$-balls w.r.t. the $\ell_p$-norms, though other constraint sets have been used too (Wong et al., 2019b). This goal can be formulated as a robust optimization problem where the objective is to minimize the adversarial population loss given some perturbation constraint $\Delta$:

$$\min_w \mathbb{E}_{(X,Y)\sim\mathcal{D}}\big[ \max_{\delta\in\Delta} \ell(X + \delta, Y; w)\big] \tag{2}$$

We have an interplay between two goals: 1) the inner-max term looks for the worst-case loss around the input, while 2) the outer-min term optimizes the hypothesis by minimizing such a loss.

Note that solving the inner-max problem is often computationally difficult, so one may approximate it with a surrogate loss obtained from a particular attack. Adversarial training and its variants (Szegedy et al., 2013; Goodfellow et al., 2014; Kurakin et al., 2016; Madry et al., 2018; Wong et al., 2019a) can be intuitively understood as an approximation of this min-max problem via different $\delta(x)$.

As shown in Section 2, one can design a regularizer that provides the **exact** value of the loss function at the attacked point for a particular choice of model, loss function, and adversary, cf. (1). Under the robust optimization framework, the regularizer and adversarial training are two realizations of the inner-max objective in (2), but using such a regularizer relieved us from using a separate inner optimization procedure, as is done in adversarial training. Motivated by that example and the robust optimization framework discussed here, we develop a regularizer that can be understood as an upper-bound on the **worst-case** value of the loss at an attacked point under a second-order approximation of the loss function.

## 4 SECOND-ORDER ADVERSARIAL REGULARIZER (SOAR)

The main idea of SOAR is to approximate the loss function using the second-order Taylor series expansion around an input $x$ and then solve the inner maximization term of the robust optimization formulation (2) using the approximated form. We show this for both $\ell_2$ and $\ell_\infty$ attacks; the same idea can be applied to other $\ell_p$ norms. We describe crucial steps of the derivation in this section, and defer details to Appendix B .

Assuming that the loss is twice-differentiable, we can approximate the loss function around input $x$ by the second-order Taylor expansion

$$\ell(x + \delta, y; w) \approx \tilde{\ell}_{2\text{nd}}(x + \delta, y; w) \triangleq \ell(x, y; w) + \nabla_x \ell(x, y; w)^\top \delta + \frac{1}{2}\delta^\top \nabla_x^2 \ell(x, y; w)\delta. \tag{3}$$

For brevity, we drop $w, y$ and use $\nabla$ to denote $\nabla_x$. Let us focus on the $\ell_p$ attacks, where the constraint set in (2) is $\Delta = \{\delta : \|\delta\|_p \leq \varepsilon\}$ for some $\varepsilon > 0$ and $p \geq 1$. We focus on the $\ell_\infty$ attack because of its popularity, but we also derive the formulation for the $\ell_2$ attacks.

As a warm-up, let us solve the inner optimization problem by considering the first-order Taylor series expansion. We have

$$\ell_{\text{FOAR}}(x) \triangleq \max_{\|\delta\|_\infty \leq \varepsilon} \ell(x) + \nabla \ell(x)^\top \delta = \ell(x) + \varepsilon \|\nabla \ell(x)\|_1. \tag{4}$$

The term $\varepsilon \|\nabla \ell(x)\|_1$ defines the First-Order Adversarial Regularizer (FOAR). This is similar to the regularizer introduced by Simon-Gabriel et al. (2019) with the choice of $\ell_\infty$ perturbation set. For a general $\ell_p$-attach with $1 \leq p \leq \infty$, we have $\|\nabla \ell(x)\|_q$ with $q$ satisfying $p^{-1} + q^{-1} = 1$. We shall empirically evaluate FOAR-based approach (for the $\ell_\infty$ attack), but our focus is going to be on solving the inner maximization problem based on the second-order Taylor expansion:

$$\max_{\|\delta\|_p \leq \varepsilon} \ell(x) + \nabla \ell(x)^\top \delta + \frac{1}{2} \delta^\top \nabla^2 \ell(x) \delta, \tag{5}$$

for $p = 2, \infty$. The second-order expansion in (3) can be rewritten as

$$\ell(x + \delta) \approx \ell(x) + \frac{1}{2} \begin{bmatrix} \delta \\ 1 \end{bmatrix}^\top \begin{bmatrix} \nabla^2 \ell(x) & \nabla \ell(x) \\ \nabla \ell(x)^\top & 1 \end{bmatrix} \begin{bmatrix} \delta \\ 1 \end{bmatrix} - \frac{1}{2} = \ell(x) + \frac{1}{2} \delta'^\top \mathbf{H} \delta' - \frac{1}{2}, \tag{6}$$

where $\delta' = [\delta; 1]$. This allows us to derive an upper bound on the expansion terms using the characteristics of a single Hessian term $\mathbf{H}$. Note that $\delta'$ is a $d + 1$-dimensional vector and $\mathbf{H}$ is a $(d + 1) \times (d + 1)$ matrix. We need to find an upper bound on $\delta'^\top \mathbf{H} \delta'$ under the attack constraint.

For the $\ell_\infty$ attack, solving this maximizing problem is not as easy as in (4) since the Boolean quadratic programming problem in formulation (5) is NP-hard. But we can relax the constraint set and find an upper bound for the maximizer. Note that with $\delta \in \mathbb{R}^d$, an $\ell_\infty$-ball of size $\varepsilon$ is enclosed by an $\ell_2$-ball of size $\sqrt{d}\varepsilon$ with the same centre. Therefore, we can upper bound the inner maximization by

$$\max_{\|\delta\|_\infty \leq \varepsilon} \ell(x + \delta) \leq \max_{\|\delta\|_2 \leq \sqrt{d}\varepsilon} \ell(x + \delta), \tag{7}$$

which after substituting the second-order Taylor series expansion leads to an $\ell_2$-constrained quadratic optimization problem

$$\ell(x) + \frac{1}{2} \max_{\|\delta\|_2 \leq \sqrt{d}\varepsilon} \delta'^\top \mathbf{H} \delta' - \frac{1}{2}, \tag{8}$$

with $\delta' = [\delta; 1]$ as before. The $\ell_2$ version of SOAR does not require this extra step, and we have $\varepsilon$ instead of $\sqrt{d}\varepsilon$ in (8). A more detailed discussion on the above relaxation procedure is included in Appendix B.2 .

**Proposition 1.** *Let $\ell : \mathbb{R}^d \to \mathbb{R}$ be a twice-differentiable function. For any $\varepsilon > 0$, we have*

$$\max_{\|\delta\|_\infty \leq \varepsilon} \tilde{\ell}_{2nd}(x + \delta) \leq \ell(x) + \frac{d\varepsilon^2 + 1}{2} \mathbb{E}\left[\|\mathbf{H}z\|_2\right] - \frac{1}{2}, \tag{9}$$

*where $\mathbf{H}$ is defined in (6) and $z \sim \mathcal{N}(0, \mathbf{I}_{(d+1) \times (d+1)})$.*

This result upper bounds the maximum of the second-order approximation $\tilde{\ell}_{2nd}$ over an $\ell_\infty$ ball with radius $\varepsilon$, and relates it to an expectation of a Hessian-vector product. Note that there is a simple correspondence between (1) and regularized loss in (9). The latter can be understood as an upper bound on the worst-case damage of an adversary under a second-order approximation of the loss. For the $\ell_2$ attack, the same line of argument leads to $\varepsilon^2 + 1$ instead of $d\varepsilon^2 + 1$.

Let us take a closer look at $\mathbf{H}z$. By decomposing $z = [z_d, z_1]^\top$, we get

$$\mathbf{H}z = \begin{bmatrix} \nabla^2 \ell(x) z_d + z_1 \nabla \ell(x) \\ \nabla \ell(x)^\top z_d + z_1 \end{bmatrix}.$$

The term $\nabla^2 \ell(x) z_d$ can be computed using Finite Difference (FD) approximation. Note that $\mathbb{E}\left[\|z_d\|_2\right] = \sqrt{d}$ for our Normally distributed $z$. To ensure that the approximation direction has the same magnitude, we use the normalized $\tilde{z}_d = \frac{z_d}{\|z_d\|_2}$ instead, and use the approximation below

$$\nabla^2 \ell(x) z_d \approx \|z_d\|_2 \frac{\nabla \ell(x + h\tilde{z}_d) - \nabla \ell(x)}{h}. \tag{10}$$

To summarize, SOAR regularizer evaluated at $x$, with a direction $z$, and FD step size $h > 0$ is

$$R(x; z, h) = \frac{d\varepsilon^2 + 1}{2} \left\| \begin{bmatrix} \|z_d\|_2 \frac{\nabla \ell(x + h\tilde{z}_d) - \nabla \ell(x)}{h} + z_1 \nabla \ell(x) \\ \nabla \ell(x)^\top z_d + z_1 \end{bmatrix} \right\|_2. \tag{11}$$

The expectation in (9) can then be approximated by taking multiple samples of $z$ drawn from $z \sim \mathcal{N}(0, \mathbf{I}_{(d+1) \times (d+1)})$. These samples would be concentrated around its expectation. One can show that $\mathbb{P}\left\{ \|Hz\| - \mathbb{E}\left[\|Hz\|\right] > t \right\} \le 2\exp(-\frac{ct^2}{\|H\|_2})$, where $c$ is a constant and $\|H\|_2$ is the $\ell_2$-induced norm (see Theorem 6.3.2 of Vershynin 2018). In practice, we observed that taking more than one sample of $z$ do not provide significant improvement for increasing adversarial robustness, and we include an empirical study on the the effect of sample sizes in Appendix E.4 .

Before we discuss the remaining details, recall that we fully robustify the model with an appropriate regularizer in Section 2. Note the maximizer of the loss based on formulation (2) is exactly the FGSM direction, and (1) shows the population loss with our FGSM-like choice of $\Delta x$. To further motivate a second-order approach, note that we can obtain the first two terms in (1) with a first-order regularizer such as FOAR; and we recover the exact form with a second-order formulation in (5).

Next, we study SOAR in the simple logistic regression setting, which shows potential failure of the regularizer and reveals why we might observe gradient masking. Based on that insight, we provide the remaining details of the method afterwards in Section 4.1.

## 4.1 Avoiding Gradient Masking

Consider a linear classifier $f : \mathbb{R}^d \times \mathbb{R}^d \to \mathbb{R}$ with $f(x; w) = \phi(\langle w, x \rangle)$, where $x, w \in \mathbb{R}^d$ are the input and the weights, and $\phi(\cdot)$ is the sigmoid function. Note that the output of $f$ has the interpretation of being a Bernoulli distribution. For the cross-entropy loss function $\ell(x, y; w) = -[y \log f(x; w) + (1-y) \log(1 - f(x; w))]$, the gradient w.r.t. the input $x$ is $\nabla \ell(x) = (f(x; w) - y)w$ and the Hessian w.r.t. the input $x$ is $\nabla^2 \ell(x) = f(x; w)(1 - f(x; w))ww^\top$.

The second-order Taylor series expansion (3) with the gradient and Hessian evaluated at $x$ is

$$\ell(x + \delta) \approx \ell(x) + r(x, y; w)w^\top \delta + \frac{1}{2}u(x; w)\delta^\top ww^\top \delta, \tag{12}$$

where $r = r(x, y; w) = f(x; w) - y$ is the residual term describing the difference between the predicted probability and the correct label, and $u = u(x; w) = f(x; w)(1 - f(x; w))$. Note that $u$ can be interpreted as how confident the model is about its predication (correct or incorrect), and is close to $0$ whenever the classifier is predicting a value close to $0$ or $1$. With this linear model, the maximization (8) becomes

$$\ell(x) + \max_{\|\delta\|_2 \le \sqrt{d}\varepsilon} \left[ rw^\top \delta + \frac{1}{2}u\delta^\top ww^\top \delta \right] = \ell(x) + \varepsilon\sqrt{d}|r(x, y; w)| \|w\|_2 + \frac{d\varepsilon^2}{2}u(x; w)\|w\|_2^2.$$

The regularization term is encouraging the norm of $w$ to be small, weighted according to the residual $r(x, y; w)$ and the uncertainty $u(x; w)$.

Consider a linear interpolation of the cross-entroply loss from $x$ to a perturbed input $x'$. Specifically, we consider $\ell(\alpha x + (1 - \alpha)x')$ for $\alpha \in [0, 1]$. Previous work has empirically shown that the value of the loss behaves logistically as $\alpha$ increases from $0$ to $1$ (Madry et al., 2018). In such a case, since there is very little curvature at $x$, if we use Hessian exactly at $x$, it leads to an inaccurate approximation of the value at $\ell(x')$. Consequently, we have a poor approximation of the inner-max, and the derived regularization will not be effective.

For the approximation in (12), this issue corresponds to the scenario in which the classifier is very confident about the clean input at $x$. Standard training techniques such as minimizing the cross-entropy loss optimize the model such that it returns the correct label with a high confidence. Whenever

---

**Algorithm 1:** Computing the SOAR objective for a single training data

**Input** : A pair of training data $(x, y)$, $\ell_\infty$ constraint of $\varepsilon$, Finite difference step-size $h$.

1 $x' \leftarrow x + \eta$, where $\eta \leftarrow (\eta_1, \eta_2, \ldots, \eta_d)^\top$ and $\eta_i \sim \mathcal{U}(-\frac{\varepsilon}{2}, \frac{\varepsilon}{2})$.

2 $x' \leftarrow \Pi_{B(x, \frac{\varepsilon}{2})} \left\{ x' + \frac{\varepsilon}{2} \text{sign}\left(\nabla_x \ell(x')\right) \right\}$ where $\Pi$ is the projection operator.

3 Sample $z \sim \mathcal{N}(0, \mathbf{I}_{(d+1)\times(d+1)})$.

4 Compute SOAR regularizer $R(x'; z, h)$ as (11).

5 Compute the pointwise objective: $\ell_{\text{SOAR}}(x, y) = \ell(x', y) + R(x'; z, h)$.

---

the classifier is correct with a high confidence, both $r$ and $u$ will be close to zero. As a result, the effect of the regularizer diminishes, i.e., the weights are no longer regularized. In such a case, the Taylor series expansion, computed using the gradient and Hessian evaluated at $x$, becomes an inaccurate approximation to the loss, and hence its maximizer is not a good solution to the inner maximization problem.

Note that this does not mean that one cannot use Taylor series expansion to approximate the loss. In fact, by the mean value theorem, there exists an $h^\star \in (0, 1)$ such that the second-order Taylor expansion is exact: $\ell(x + \delta) = \ell(x) + \nabla\ell(x)^\top \delta + \frac{1}{2}\delta^\top \nabla^2 \ell(x + h^\star \delta)\delta$. The issue is that if we compute the Hessian at $x$ (instead of at $x + h^\star \delta$), our approximation might not be very good whenever the curvature profile of the loss function at $x$ is drastically different from the one at $x + h^\star \delta$.

More importantly, a method relying on the gradient masking can be easily circumvented (Athalye et al., 2018). Our early experimental results had also indicated that gradient masking occurred with SOAR when the gradient and Hessian were evaluated at $x$. In particular, we observe that SOAR with zero-initialization leads to models with nearly 100% confidence on their predictions, leading to an ineffective regularizer. The result is reported in Table 5 in Appendix D.

This suggests a heuristic to improve the quality of SOAR. That is to evaluate the gradient and Hessian, through FD approximation (10) at a less confident point in the $\ell_\infty$ ball of $x$. We found that evaluating the gradient and Hessian at 1-step PGD adversary successfully circumvent the issue (Line 1-2 in Algorithm 1). We compare other initializations in Table 5 in Appendix D. To ensure the regularization is of the original $\ell_\infty$ ball of $\varepsilon$, we use $\frac{\varepsilon}{2}$ for PGD1 initialization, and then $\frac{\varepsilon}{2}$ in SOAR.

Based on this heuristic, the regularized pointwise objective for a data point $(x, y)$ is

$$\ell_{\text{SOAR}}(x, y) = \ell(x', y) + R(x'; z, h), \tag{13}$$

where $z \sim \mathcal{N}(0, \mathbf{I}_{(d+1)\times(d+1)})$ and the point $x'$ is initialized at PGD1 adversary. Algorithm 1 summarizes SOAR on a single training data. We include the full training procedure in Appendix C . Moreover, we include additional discussions and experiments on gradient masking in Appendix E.11.

## 4.2 RELATED WORK

Several regularization-based alternatives to adversarial training have been proposed. Simon-Gabriel et al. (2019) studied regularization under the first-order Taylor approximation. The proposed regularizer for the $\ell_\infty$ perturbation set is the same as FOAR. Qin et al. (2019) propose local linearity regularization (LLR), where the local linearity measure is defined by the maximum error of the first-order Taylor approximation of the loss. LLR minimizes the local linearity mesaure, and minimizes the magnitude of the projection of gradient along the corresponding direction of the local linearity mesaure. It is motivated by the observation of flat loss surfaces during adversarial training.

CURE (Moosavi-Dezfooli et al., 2019) is the closest to our method. They empirically observed that adversarial training leads to a reduction in the magnitude of eigenvalues of the Hessian w.r.t. the input. Thus, they proposed directly minimizing the curvature of the loss function to mimic the effect of adversarial training. An important advantage of our proposed method is that SOAR is derived from a complete second-order Taylor approximation of the loss, while CURE exclusively focuses on the second-order term for the estimation of the curvature. Note the final optimization objective in SOAR, FOAR, LLR and CURE contains derivative w.r.t. the input of the DNN, and such a technique was first introduced to improve generalization by Drucker & Le Cun (1992) as double backpropagation.

Table 1: Performance on CIFAR-10 against $\ell_\infty$ bounded white-box PGD attacks ($\varepsilon = 8/255$).

| | Method | Standard | FGSM | PGD20 | PGD100 | PGD200 | PGD1000 | PGD20-50 |
|---|---|---|---|---|---|---|---|---|
| **WideResNet** | Standard | **95.79**% | 44.77% | 0.03% | 0.01% | 0.00% | 0.00% | 0.00% |
| | ADV | 87.14% | 55.38% | 45.64% | 45.05% | 45.01% | 44.95% | 45.21% |
| | TRADES | 84.92% | 60.87% | **55.40**% | **55.10**% | **55.11**% | **55.06**% | **55.05**% |
| | MART | 83.62% | **61.61**% | **56.29**% | **56.11**% | **56.10**% | **56.07**% | **55.98**% |
| | MMA | 84.36% | **61.97**% | 52.01% | 51.26% | 51.28% | 51.02% | 50.93% |
| **ResNet** | Standard | **92.54**% | 21.59% | 0.14% | 0.09% | 0.10% | 0.08% | 0.10% |
| | ADV | 80.64% | 50.96% | 42.86% | 42.27% | 42.21% | 42.17% | 42.55% |
| | TRADES | 75.61% | 50.06% | 45.38% | 45.19% | 45.18% | 45.16% | 45.24% |
| | MART | 75.88% | 52.55% | 46.60% | 46.29% | 46.25% | 46.21% | 46.40% |
| | MMA | 82.37% | 47.08% | 37.26% | 36.71% | 36.66% | 36.64% | 36.85% |
| | FOAR | 65.84% | 36.96% | 32.28% | 31.87% | 31.89% | 31.89% | 32.08% |
| | SOAR | 87.95% | **67.15**% | **56.06**% | **55.00**% | **54.94**% | **54.69**% | **54.20**% |

Another related line of adversarial regularization methods do not involve approximation to the loss function nor robust optimization. TRADES (Zhang et al., 2019) introduces a regularization term that penalizes the difference between the output of the model on a training data and its corresponding adversarial example. MART (Wang et al., 2020) reformulated the training objective by explicitly differentiating between the mis-classified and correctly classified examples. Ding et al. (2019b) present another regularization approach that leverages adaptive margin maximization (MMA) on correctly classified example to robustify the model.

# 5 EXPERIMENTS

In this section, we verify the effectiveness of the proposed regularization method against $\ell_\infty$ PGD attacks on CIFAR10. Our experiments show that training with SOAR leads to significant improvements in adversarial robustness, which is achieved without significantly sacrificing standard accuracy. We focus on $\ell_\infty$ in this section and defer evaluations on $\ell_2$ in Appendix E.5. Additionally, we provide a detailed discussion and evaluations on the SVHN dataset in Appendix E.6.

We train ResNet-10 (He et al., 2016) on the CIFAR-10 dataset. The baseline methods consist of: (1) Standard: training with no adversarially perturbed data; (2) ADV: training with 10-step PGD adversarial examples; (3) TRADES; (4) MART and (5) MMA. Empirical studies in Madry et al. (2018) and Wang et al. (2020) reveal that their approaches benefit from increasing model capacity to achieve higher adversarial robustness, as such, we include WideResNet (Zagoruyko & Komodakis, 2016) for all baseline methods. We were not able to reproduce the results of two closely related works, CURE and LLR, which we discuss further in Appendix E.1. In Appendix E.13, we compare SOAR and FOAR with different initializations. FOAR achieves the best adversarial robustness using PGD1 initialization, so we only present this variation of FOAR in this section.

The optimization procedure is described in detail in Appendix E.2. Note that all methods in this section are trained to defend against $\ell_\infty$ norm attacks with $\varepsilon = 8/255$, as this is a popular choice of $\varepsilon$ in the literature. The PGD adversaries discussed in Sections 5.1 and 5.2 are generated with $\varepsilon = 8/255$ and a step size of $2/255$ (pixel values are normalized to $[0, 1]$). PGD20-50 denotes 20-step PGD attacks with 50 restarts. In Section 5.3, we compare SOAR with baseline methods on $\ell_\infty$ AutoAttack (Croce & Hein, 2020) adversaries with a varying $\varepsilon$. Additionally, results of all methods on ResNet10 are obtained by averaging over 3 independently initialized and trained models, where the standard deviations are reported in Appendix E.10. We use the provided pretrained WideResNet model provided in the public repository of each method. Lastly, discussions on challenges (i.e., difficult to train from scratch, catastrophic overfitting, BatchNorm, etc.) we encountered while implementing SOAR and our solutions (i.e., using pretrained model, clipping regularizer gradient, early stopping, etc.) are included in Appendix E.7.

Table 2: Performance on CIFAR-10 against $\ell_\infty$ bounded black-box attacks ($\varepsilon = 8/255$).

|  | Method | SimBA | PGD20-R | PGD20-W | PGD1000-R | PGD1000-W |
|---|---|---|---|---|---|---|
| WideResNet | ADV | 49.20% | **84.69**% | **86.30**% | **84.69**% | **86.24**% |
| | TRADES | 58.97% | 82.18% | 83.88% | 82.24% | 83.95% |
| | MART | 59.60% | 80.79% | 82.62% | 80.96% | 82.76% |
| | MMA | 58.30% | 80.73% | 82.74% | 80.76% | 82.74% |
| ResNet | ADV | 47.27% | 77.19% | 79.48% | 77.22% | 79.55% |
| | TRADES | 47.67% | 72.28% | 74.39% | 72.24% | 74.37% |
| | MART | 48.57% | 72.99% | 74.91% | 72.99% | 75.04% |
| | MMA | 43.53% | 78.70% | 80.39% | 78.72% | 81.35% |
| | FOAR | 35.97% | 63.56% | 65.20% | 63.60% | 65.27% |
| | SOAR | **68.57**% | **79.25**% | **86.35**% | **79.49**% | **86.47**% |

## 5.1 ROBUSTNESS AGAINST PGD WHITE-BOX ATTACKS

Before making the comparison between SOAR and the baselines in Table 1, note that FOAR achieves 32.28% against PGD20 attacks. Despite its uncompetitive performance, this shows that approximating the robust optimization formulation based on Taylor series expansion is a reasonable approach. Furthermore, this justifies our extension to a second-order approximation, as the first-order alone is not sufficient. Lastly, we observe that training with SOAR significantly improves the adversarial robustness against all PGD attacks, leading to higher robustness in all k-step PGD attacks on the ResNet model. SOAR remains competitive compared to baseline methods trained on high-capacity WideResNet architecture.

## 5.2 ROBUSTNESS AGAINST BLACK-BOX ATTACKS

Many defences only reach an illusion of robustness through methods collectively known as gradient masking (Athalye et al., 2018). These methods often fail against attacks generated from an undefended independently trained model, known as transfer-based black-box attacks. Recent works (Tramèr et al., 2017; Ilyas et al., 2019) have proposed hypotheses for the success of transfer-based black-box attacks. In our evaluation, the transferred attacks are PGD20 and PGD1000 perturbations generated from two source models: ResNet and WideResNet, which are denoted by the suffix -R and -W respectively. The source models are trained separately from the defence models on the unperturbed training set. Additionally, Tramer et al. (2020) recommends score-based black-box attacks such as SimBA (Guo et al., 2019). They are more relevant in real-world applications where gradient information is not accessible, and are empirically shown to be more effective than transfer-based attacks. Because they are solely based on the confidence score of the model, score-based attacks are resistant to gradient-masking. All black-box attacks in this section are $\ell_\infty$ constrained at $\varepsilon = 8/255$.

SOAR achieves the best robustness against all baseline methods trained on ResNet, as shown in Table 2. Compared with the baselines trained on WideResNet, SOAR remains the most robust model against transferred PGD20-W and PGD1000-W, approaching its standard accuracy on unperturbed data. Note that all defence methods are substantially more vulnerable to the score-based SimBA attack. SOAR regularized model is the most robust method against SimBA.

## 5.3 ROBUSTNESS AGAINST AUTOATTACK

During the ICLR rebuttal phase, we evaluated SOAR against Autoattack (Croce & Hein, 2020). In this section, we focus on the $\ell_\infty$-bounded Autoattack, and similar results with the $\ell_2$-bounded attack is included in Appendix E.5. We noticed that SOAR has shown greater vulnerabilities to AutoAttack compared to the attacks discussed in Sections 5.1 and 5.2. AutoAttack consists of an ensemble of four attacks: two parameter-free versions of the PGD attack (APGD-CE and APGD-DLR), a white-box fast adaptive boundary (FAB) attack (Croce & Hein, 2019), a score-based black-box Square Attack (Andriushchenko et al., 2020). Notice that the major difference between the two PGD attacks is the

Table 3: Performance of the ResNet models on CIFAR-10 against the four $\ell_\infty$ bounded attacks used as an ensemble in AutoAttack ($\varepsilon = 8/255, 6/255, 4/255$).

| Method | Untargeted APGD-CE | Targeted APGD-DLR | Targeted FAB | Square Attack |
|--------|--------------------|--------------------|--------------|---------------|
| ADV | 41.57\|51.84\|61.95% | 38.99\|**49.65**\|**60.11**% | 39.68\|**50.05**\|**60.26**% | **47.84**\|**56.09**\|63.02% |
| TRADES | 44.69\|53.11\|60.67% | **40.27**\|49.08\|58.25% | **40.64**\|49.39\|58.45% | 46.16\|54.04\|61.22% |
| MART | 45.01\|53.52\|62.05% | 39.22\|48.48\|58.38% | 39.90\|49.11\|58.65% | 46.90\|54.54\|62.03% |
| MMA | 35.59\|46.31\|58.15% | 34.77\|45.90\|57.82% | 35.50\|46.54\|58.24% | 45.24\|54.05\|63.99% |
| FOAR | 31.15\|40.16\|49.87% | 27.56\|36.92\|46.91% | 27.92\|37.21\|47.04% | 35.92\|43.18\|51.05% |
| SOAR | **53.40**\|**58.34**\|**63.87**% | 18.25\|33.22\|52.64% | 20.22\|34.04\|53.29% | 35.94\|49.65\|**63.90**% |

loss they are based on: APGD-CE is based on the cross-entropy loss similar to (Madry et al., 2018), and APGD-DLR is based on the logit difference similar to (Carlini & Wagner, 2017).

To better understand the source of SOAR's vulnerability, we tested it against the four attacks individually. First, we observed that the result against untargeted APGD-CE is similar to the one shown in Section 5.1. This is expected because the attacks are both formulated based on cross-entropy-based PGD. However, there is a considerable degradation in the accuracy of SOAR against targeted APGD-DLR and targeted FAB. At $\varepsilon = 8/255$, SOAR is most vulnerable to targeted APGD-DLR with a robust accuracy of only $18.25\%$. To further investigate SOAR's robustness against AutoAttack, we tested with different $\varepsilon$ to verify if SOAR can at least improve robustness against $\ell_\infty$ attacks with smaller $\varepsilon$. We observed that at $\varepsilon = 4/255$ the robustness improvement of SOAR becomes more consistent. Interestingly, we also noticed that a model with better robustness at $\varepsilon = 8/255$ does not guarantee a better robustness at $\varepsilon = 4/255$, as is the case for Square Attack on ADV and SOAR.

Combing the results with the four attacks and with different $\varepsilon$, we provide three hypotheses on the vulnerability of SOAR. First, SOAR might overfit to a particular type of attack: adversarial examples generated based on the cross-entropy loss. APGD-DLR is based on logit difference and FAB is based on finding minimal perturbation distances, which are both very different from the cross-entropy loss. Second, SOAR might rely on gradient masking *to a certain extent*, and thus PGD with cross-entropy loss is difficult to find adversaries while they still exist. This also suggests that the results with black-box attacks might be insufficient to conclusively eliminate the possibility of gradient masking. Third, since SOAR provide a more consistent robustness improvement at a smaller $\varepsilon$, this suggests that the techniques discussed in Section 4 did not completely address the problems raised from the second-order approximation. This makes the upper-bound of the inner-max problem loose, hence making SOAR improves robustness against attacks with $\varepsilon$ smaller than what it was formulated with.

Finally, we emphasize that this should not rule SOAR as a failed defence. Previous work shows that a mechanism based on gradient masking can be *completely* circumvented, resulting in a $0\%$ accuracy against non-gradient-based attacks (Athalye et al., 2018). Our result on SimBA and Square Attack shows that this is not the case with SOAR, even at $\varepsilon = 8/255$, and thus the robustness improvement cannot be *only* due to gradient masking. Overall, we think SOAR's vulnerability to AutoAttack is an interesting observation and requires further investigation.

## 6  CONCLUSION

This work proposed SOAR, a regularizer that improves the robustness of DNN to adversarial examples. SOAR was obtained using the second-order Taylor series approximation of the loss function w.r.t. the input, and approximately solving the inner maximization of the robust optimization formulation. We showed that training with SOAR leads to significant improvement in adversarial robustness under $\ell_\infty$ and $\ell_2$ attacks. This is only one step in designing better regularizers to improve the adversarial robustness. Several directions deserve further study, with the prominent one being SOAR's vulnerabilities to AutoAttack. Another future direction is to understand the loss surface of DNN better in order to select a good point around which an accurate Taylor approximation can be made. This is important for designing regularizers that are not affected by gradient masking.

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

# A    DERIVATIONS OF SECTION 2: LINEAR REGRESSION WITH AN OVER-PARAMETRIZED MODEL

We derive the results reported in Section 2 in more detail here. Recall that we consider a linear model $f_w(x) = \langle w, x \rangle$ with $x, w \in \mathbb{R}^d$. We suppose that $w^* = (1, 0, 0, \ldots, 0)^\top$ and the distribution of $x \sim p$ is such that it is confined on a 1-dimensional subspace $\{ (x_1, 0, 0, \ldots, 0) : x_1 \in \mathbb{R} \}$. So the density of $x$ is $p((x_1, \ldots, x_d)) = p_1(x_1)\delta(x_2)\delta(x_3)\ldots\delta(x_d)$, where $\delta(\cdot)$ is Dirac's delta function.

We initialize the weights at the first time step as $w(0) \sim N(0, \sigma^2 \mathbf{I}_{d \times d})$, and use GD to find the minimizer of the population loss. The partial derivatives of the population loss are

$$\frac{\partial \mathcal{L}(w)}{\partial w_j} = \begin{cases} \int (w_1 - w_1^*)p_1(x_1)x\mathrm{d}x = (w_1 - w_1^*)\mu_1, \\ \int (w_j - w_j^*)\delta(x_j)x\mathrm{d}x = (w_j - w_j^*)0 = 0, \ j \neq 1. \end{cases}$$

where $\mu_1 = \mathbb{E}[X_1]$. Notice that the gradient in dimension $j = 1$ is non-zero, unless $(w_1 - w_1^*)\mu_1 = 0$. Assuming that $\mu_1 \neq 0$, this implies that the gradient won't be zero unless $w_1 = w_1^*$. On the other hand, the gradients in dimensions $j = 2, \ldots, d$ are all zero, so GD does not change the value of $w_j(t)$ for $j = 2, \ldots, d$. Therefore, under the proper choice of learning rate $\beta$, we get that the asymptotic solution of GD solution is $\bar{w} \triangleq \lim_{r \to \infty} w(t) = (w_1^*, w_2(0), w_3(0), \ldots, w_d(0))^\top$. It is clear that $\mathcal{L}(\bar{w}) = 0$, i.e., the population loss is zero, as noted already as our first observation in that section.

Also note that we can easily attack this model by perturbing $x$ by $\Delta x = (0, \Delta x_2, \Delta x_3, \ldots, \Delta x_d)^\top$. The pointwise loss at $x + \Delta x$ is

$$l(x + \Delta x; w) = \frac{1}{2}|(w_1 - w_1^*)x_1 + \langle w, \Delta x \rangle|^2 = \frac{1}{2}|r(x; w) + \langle w, \Delta x \rangle|^2.$$

With the choice of $\Delta x_i = \varepsilon \operatorname{sign}(w_i(0))$ (for $i = 2, \ldots, d$) and $\Delta x_1 = 0$, an FGSM-like attack (Goodfellow et al., 2014) at the learned weight $\bar{w}$ leads to the pointwise loss of

$$l(x + \Delta x; \bar{w}) = \frac{1}{2}\varepsilon^2 \left[ \sum_{j=2}^d |w_j(0)| \right]^2 \approx \frac{1}{2}\varepsilon^2 \|w(0)\|_1^2.$$

We comment that our choice of $\Delta x$ is not from the same distribution as the training data $x$. This choice aligns with the hypotheses in Ding et al. (2019a); Schmidt et al. (2018) that adversarial examples come from a shifted data distribution; however, techniques such as feature adversaries (Sabour et al., 2015) focus on designing perturbations to be close to input distributions. We stress that the goal here is to illustrate the loss under this particular attack.

In order to get a better sense of this loss, we compute its expected value w.r.t. the randomness of weight initialization. We have that (including the extra $|w_1(0)|$ term too)

$$\mathbb{E}_{W \sim N(0, \sigma^2 \mathbf{I}_{d \times d})} \left[ \|W\|_1^2 \right] = \mathbb{E} \left[ \sum_{i,j=1}^d |W_i||W_j| \right] = \sum_{i=1}^d \mathbb{E}\left[|W_i|^2\right] + \sum_{i,j=1, i \neq j}^d \mathbb{E}[|W_i|]\,\mathbb{E}[|W_j|],$$

where we used the independence of the r.v. $W_i$ and $W_j$ when $i \neq j$. The expectation $\mathbb{E}\left[|W_i|^2\right]$ is the variance $\sigma^2$ of $W_i$. The r.v. $|W_j|$ has a folded normal distribution, and its expectation $\mathbb{E}[|W_j|]$ is $\sqrt{\frac{2}{\pi}}\sigma$. Thus, we get that

$$\mathbb{E}_{W \sim N(0, \sigma^2 \mathbf{I}_{d \times 1})} \left[ \|W\|_1^2 \right] = d\sigma^2 + d(d-1)\frac{2}{\pi}\sigma^2 \approx \frac{2}{\pi}d^2\sigma^2,$$

for $d \gg 1$. The expected population loss of the specified attack $\Delta x$ at the asymptotic solution $\bar{w}$ is

$$\mathbb{E}_{X,W}\left[l(X + \Delta x); \bar{w}\right] \approx O(\varepsilon^2 d^2 \sigma^2).$$

The dependence of this loss on dimension $d$ is significant, showing that the learned model is quite vulnerable to attacks. We note that the conclusions would not change much with initial distributions other than the Normal distribution.

An effective solution is to regularize the loss to encourage the weights of irrelevant dimensions going to zero. A generic regularizer is to use the $\ell_2$-norm of the weights, i.e., formulate the problem as a ridge regression. In that case, the regularized population loss is

$$\mathcal{L}_{\text{ridge}}(w) = \frac{1}{2}\mathbb{E}\left[|\langle X, w \rangle - \langle X, w^* \rangle|^2\right] + \frac{\lambda}{2}\|w\|_2^2.$$

One can see that the solution of $\nabla_w \mathcal{L}_{\text{ridge}}(w) = 0$ is $\bar{w}_1(\lambda) = \frac{\mu_1}{\mu_1 + \lambda}w_1^*$ and $\bar{w}_j(\lambda) = 0$ for $j \neq 1$.

$$\bar{w}_j(\lambda) = \begin{cases} \frac{\mu_1}{\mu_1 + \lambda}w_1^* & j = 1 \\ 0 & j \neq 1. \end{cases}$$

The use of this generic regularizer seems reasonable in this example, as it enforces the weights for dimensions 2 to $d$ to become zero. Its only drawback is that it leads to a biased estimate of $w_1^*$. The bias, however, can be made small with a small choice for $\lambda$. We can obtain a similar conclusion for the $\ell_1$ regularizer (Lasso).

As such, one has to define a regularizer that is specially-designed for improving adversarial robustness. Bishop (1995) showed the strong connection between training with random perturbation and Tikhonov Regularization. Inspired by this idea, we develop a regularizer that mimics the adversary itself. Let us assume that a particular adversary attacks the model by adding $\Delta x = (0, \varepsilon\,\text{sign}(w_2(0)), \ldots, \varepsilon\,\text{sign}(w_d(0))^\top$. The population loss at the perturbed point is

$$\mathcal{L}_{\text{robustified}}(w) \triangleq \mathbb{E}\left[l(X + \Delta x; w)\right] = \frac{1}{2}\mathbb{E}\left[\left|r(x; w) + \varepsilon\sum_{j=2}^{d}|w_j|\right|^2\right]$$

$$= \mathcal{L}(w) + \varepsilon\mathbb{E}\left[r(X; w)\right]\|w_{2:d}\|_1 + \frac{\varepsilon^2}{2}\|w_{2:d}\|_1^2,$$

where $\|w_{2:d}\|_1 = \sum_{j=2}^{d}|w_j|$.[1] This is the same objective as (1) reported in Section 2. Note that minimizing $\mathcal{L}_{\text{robustified}}(w)$ is equivalent to minimizing the model at the point $x' = x + \Delta x$. The regularizer $\varepsilon\mathbb{E}\left[r(X; w)\right]\|w_{2:d}\|_1 + \frac{\varepsilon^2}{2}\|w_{2:d}\|_1^2$ incorporates the effect of adversary in exact form. This motivated the possibility of designing a regularized tailored to prevent attacks.

## A.1 DERIVATION OF THE POPULATION LOSS UNDER ITS FIRST AND SECOND ORDER APPRXOIMATION

First, we show that the FGSM direction is the maximizer of the loss when the perturbation is $\ell_\infty$ constrained. Based on the pointwise loss at $x + \Delta x$, we have

$$\max_{\|\Delta X\|_\infty \leq \varepsilon} l(x + \Delta x; w) = \frac{1}{2}\left|r(x; w) + \max_{\|\Delta X\|_\infty \leq \varepsilon}\langle w, \Delta x \rangle\right|^2.$$

We use the Cauchy-Schwarz inequality to obtain

$$\max_{\|\Delta X\|_\infty \leq \varepsilon}\langle w, \Delta x \rangle \leq \max_{\|\Delta X\|_\infty \leq \varepsilon}|\langle w, \Delta x \rangle| \leq \max_{\|\Delta X\|_\infty \leq \varepsilon}\|w\|_1\|\Delta x\|_\infty = \varepsilon\|w\|_1,$$

which leads to

$$\underset{\|\Delta X\|_\infty \leq \varepsilon}{\text{argmax}}\ l(x + \Delta x; w) = \varepsilon\,\text{sign}(w).$$

Next, we show that the first-order approximation of $\mathbb{E}\left[l(X + \Delta x; w)\right]$ obtains the first two terms in (1).

Note the gradient of the loss w.r.t. the input is

$$\nabla_x l(x; w) = (\langle w, \Delta x \rangle - \langle w^*, \Delta x \rangle)(w - w^*) = r(x; w)(w - w^*),$$

---

[1]A similar, but more complicated result, would hold if the adversary could also attack the first dimension.

and the Hessian w.r.t. the input is

$$\nabla_x^2 l(x; w) = (w - w^*)(w - w^*)^\top.$$

The first-order Taylor series approximation is

$$\begin{aligned}
\mathcal{L}_{\text{robustified}}(w) \approx \hat{\mathcal{L}}_{\text{1st}}(w) &\triangleq \mathbb{E}\left[ l(X; w) + \nabla_x l(X; w)^\top \Delta x \right] \\
&= \mathcal{L}(w) + \mathbb{E}\left[ r(X; w)(w - w^*)^\top \Delta x \right] \\
&= \mathcal{L}(w) + \mathbb{E}\left[ r(X; w) w^\top \Delta x \right] \\
&= \mathcal{L}(w) + \varepsilon \mathbb{E}\left[ r(X; w) \right] \|w_{2:d}\|_1.
\end{aligned}$$

Note that $w^{*\top} \Delta x = 0$ because of our particular choice of $\Delta x$ and $w^*$. Here we obtain the first two terms in (1).

The second-order Taylor series approximation is

$$\begin{aligned}
\mathcal{L}_{\text{robustified}}(w) \approx \hat{\mathcal{L}}_{\text{2nd}}(w) &\triangleq \mathbb{E}\left[ l(X; w) + \nabla_x l(X; w)^\top \Delta x + \frac{1}{2} \Delta x^\top \nabla_x^2 l(x; w) \Delta x \right] \\
&= \mathcal{L}(w) + \varepsilon \mathbb{E}\left[ r(X; w) \right] \|w_{2:d}\|_1 + \frac{1}{2} \Delta x^\top (w - w^*)(w - w^*)^\top \Delta x \\
&= \mathcal{L}(w) + \varepsilon \mathbb{E}\left[ r(X; w) \right] \|w_{2:d}\|_1 + \frac{\varepsilon^2}{2} \|w_{2:d}\|_1^2,
\end{aligned}$$

which recovers the exact form in (1).

This completes the motivation of using second-order Taylor series approximation with our warm-up toy example.

# B DERIVATIONS OF SECTION 4: SECOND-ORDER ADVERSARIAL REGULARIZER (SOAR)

## B.1 RELAXATION

Note the Boolean quadratic programming (BQP) problem in formulation (5) is NP-hard (Beasley, 1998; Lima & Grossmann, 2017). Even though there exist semi-definite programming (SDP) relaxations, such approaches require the exact Hessian w.r.t. the input, which is computationally expensive to obtain for high-dimensional inputs. And even if we could compute the exact Hessian, SDP itself is a computationally expensive approach, and not suitable to be within the inner loop of a DNN training. As such, we relax the $\ell_\infty$ constraint to an $\ell_2$ constraint, which as we see, leads to a computationally efficient solution.

## B.2 THE ISSUE RELATED TO THE LOOSENESS OF THE BOUND IN EQ (7)

In the ICLR rebuttal phase, the reviewer pointed out that, from the perspective of the volume ratio between the two $\ell_p$ balls, replacing $\|\delta\|_\infty \leq \varepsilon$ with $\|\delta\|_2 \leq \sqrt{d}\varepsilon$ can be problematic since the volume of $\{\delta : \|\delta\|_\infty \leq \varepsilon\}$ is $2^d \varepsilon^d$ whereas the volume of $\{\delta : \|\delta\|_2 \leq \sqrt{(d)}\varepsilon\}$ is $\frac{\pi^{d/2}}{\Gamma(1+d/2)} d^{d/2} \varepsilon^d$. Their ratio goes to 0 as the dimension increases. The implication is that the search space for the $\ell_\infty$ maximizer is infinitesimal compared to the one for the $\ell_2$ maximizer, leading to a loose upper-bound.

As a preliminary study on the tightness of the bound, we evaluated the two slides of (7) by approximating the maximum using PGD attacks. In particular, we approximate $\max_{\|\delta\|_\infty \leq \epsilon} \ell(x+\delta)$ using $\ell(x+\delta_\infty)$ where $\delta_\infty$ is generated using 20-iteration $\ell_\infty$-PGD with $\epsilon = \frac{8}{255}$. Similarly, we approximate $\max_{\|\delta\|_2 \leq \sqrt{d}\epsilon} \ell(x+\delta)$ using $\ell(x+\delta_2)$ where $\delta_2$ is generated using 100-iteration $\ell_2$-PGD with $\epsilon = 1.74$. The reason for this particular configuration of attack parameter is to match the ones used during our previous evaluations.

Table 4: Comparing $\ell(x+\delta_\infty)$ and $\ell(x+\delta_2)$. We approximate $\delta_\infty$ and $\delta_2$ using the PGD attacks with their corresponding $\ell_p$ norm.

| Checkpoints | $\ell(x+\delta_\infty)$ | $\ell(x+\delta_2)$ |
|---|---|---|
| Beginning of SOAR | 76.03 | 112.10 |
| End of SOAR | 7.0 | 20.15 |

From this preliminary study, we observe that there is indeed a gap between the approximated LHS and RHS of (7), and thus, we think it is a valuable future research direction to explore other possibilities that allow us to use a second-order approximation to study the worst-case loss subject to an constrained perturbation.

## B.3 UNIFIED OBJECTIVE

We could maximize each term inside (8) separately and upper bound the max by $\max_{\|\delta\|_2 \leq \sqrt{d}\varepsilon} \nabla\ell(x)^\top \delta + \max_{\|\delta\|_2 \leq \sqrt{d}\varepsilon} \frac{1}{2}\delta^\top \nabla^2\ell(x)\delta = \sqrt{d}\varepsilon \|\nabla\ell(x)\|_2 + \frac{1}{2}d\varepsilon^2 \sigma_{\max}(\nabla^2\ell(x))$, where $\sigma_{\max}(\nabla^2\ell(x))$ is the largest singular value of the Hessian matrix, $\nabla^2\ell(x)$. Even though the norm of the gradient and the singular value of the Hessian have an intuitive appeal, separately optimizing these terms might lead to a looser upper bound than necessary. The reason is that the maximizer of the first two terms are $\mathrm{argmax}\left|\nabla\ell(x)^\top\delta\right| = \frac{\nabla\ell(x)}{\|\nabla\ell(x)\|_2}$ and the direction corresponding to the largest singular value of $\nabla^2\ell(x)$. In general, these two directions are not aligned.

### B.4 PROOF OF PROPOSITION 1

*Proof.* By the inclusion of the $\ell_\infty$-ball of radius $\varepsilon$ within the $\ell_2$-ball of radius $\sqrt{d}\varepsilon$ and the definition of $\mathbf{H}$ in (6), we have

$$
\begin{aligned}
\max_{\|\delta\|_\infty \leq \varepsilon} \tilde{\ell}_{\text{2nd}}(x) &\leq \max_{\|\delta\|_2 \leq \sqrt{d}\varepsilon} \tilde{\ell}_{\text{2nd}}(x) \\
&= \max_{\|\delta\|_2 \leq \sqrt{d}\varepsilon} \ell(x) + \frac{1}{2} \begin{bmatrix} \delta \\ 1 \end{bmatrix}^\top \begin{bmatrix} \nabla^2\ell(x) & \nabla\ell(x) \\ \nabla\ell(x)^\top & 1 \end{bmatrix} \begin{bmatrix} \delta \\ 1 \end{bmatrix} - \frac{1}{2} \\
&= \ell(x) + \frac{1}{2} \max_{\|\delta\|_2 \leq \sqrt{d}\varepsilon} \begin{bmatrix} \delta \\ 1 \end{bmatrix}^\top \mathbf{H} \begin{bmatrix} \delta \\ 1 \end{bmatrix} - \frac{1}{2} \\
&\leq \ell(x) + \frac{1}{2} \max_{\|\delta'\|_2 \leq \sqrt{d\varepsilon^2+1}} \delta'^\top \mathbf{H} \delta' - \frac{1}{2}.
\end{aligned}
$$

It remains to upper bound $\max_{\|\delta'\|_2 \leq \varepsilon'} \delta'^\top \mathbf{H} \delta'$ with $\varepsilon' = \sqrt{d\varepsilon^2 + 1}$. We use the Cauchy-Schwarz inequality to obtain

$$
\max_{\|\delta'\|_2 \leq \varepsilon'} \delta'^\top \mathbf{H} \delta' \leq \max_{\|\delta'\|_2 \leq \varepsilon'} \left| \delta'^\top \mathbf{H} \delta' \right| \leq \max_{\|\delta'\|_2 \leq \varepsilon'} \|\delta'\|_2 \|\mathbf{H}\delta'\|_2 = \varepsilon' \max_{\|\delta'\|_2 \leq \varepsilon'} \|\mathbf{H}\delta'\|_2 = \varepsilon'^2 \|\mathbf{H}\|_2 \,,
$$

where the last equality is obtained using properties of the $\ell_2$-induced matrix norm (this is the spectral norm). Since computing $\|\mathbf{H}\|_2$ would again require the exact input Hessian, and we would like to avoid it, we further upper bound the spectral norm by the Frobenius norm as

$$
\|\mathbf{H}\|_2 = \sigma_{\max}(\mathbf{H}) \leq \|\mathbf{H}\|_{\text{F}} \,.
$$

The Frobenius norm itself satisfies

$$
\|\mathbf{H}\|_{\text{F}} = \sqrt{\text{Tr}(\mathbf{H}^\top \mathbf{H})} = \mathbb{E}\left[ \|\mathbf{H}z\|_2 \right], \tag{14}
$$

where $z \sim \mathcal{N}(0, \mathbf{I}_{(d+1)\times(d+1)})$. Therefore, we can estimate $\|\mathbf{H}\|_{\text{F}}$ by sampling random vectors $z$ and compute the sample average of $\|\mathbf{H}z\|_2$. $\qquad\square$

## C    SOAR ALGROITHM: A COMPLETE ILLUSTRATION

In Algorithm 1, we present the inner-loop operation of SOAR using a single data point. Here we summarize the full training procedure with SOAR in Algorithm 2. Note that it is presented as if the optimizer is SGD, but we may use other optimizers as well.

---

**Algorithm 2:** Improving adversarial robustness via SOAR

**Input**  :Training dataset. Learning rate $\beta$, training batch size $b$, number of iterations $N$, $\ell_\infty$ constraint of $\varepsilon$, Finite difference step-size $h$.

1  Initialize network with pre-trained weight $w$;
2  **for** $i \in \{0, 1, \ldots, N\}$ **do**
3      Get mini-batch $B = \{(x_1, y_1), \cdots, (x_b, y_b)\}$ from the training set.
4      **for** $j = 1, \ldots, m$ *(in parallel)* **do**
5          $x'_j \leftarrow x_j + \eta$, where $\eta \leftarrow (\eta_1, \eta_2, \ldots, \eta_d)^\top$ and $\eta_i \sim \mathcal{U}(-\frac{\varepsilon}{2}, \frac{\varepsilon}{2})$.
6          $x'_j \leftarrow \Pi_{B\left(x_j, \frac{\varepsilon}{2}\right)} \left\{ x'_j + \frac{\varepsilon}{2} \operatorname{sign} \left( \nabla_{x'_j} \ell(x'_j) \right) \right\}$ where $\Pi$ is the projection operator.
7          Sample $z \sim \mathcal{N}(0, \mathbf{I}_{(d+1) \times (d+1)})$.
8          Compute the SOAR regularizer $R(x'_j; z, h)$ as (11).
9          Compute the pointwise objective: $\ell_{\text{SOAR}}(x_j, y_j) = \ell(x'_j, y_j) + R(x'_j; z, h)$.
10     **end**
11     $w_{i+1} \leftarrow w_i - \beta \times \frac{1}{b} \sum_{j=1}^b \nabla_{w_i} \ell_{\text{SOAR}}$.
12  **end**

---

# D    POTENTIAL CAUSES OF GRADIENT MASKING

Table 5: Average value of the highest probability output for all test set data, that is, $\frac{1}{N}\Sigma_{n=1}^{N}\max_{i\in 1,2,\ldots,c}P(x_n)_i$, where $P(x_n)_i$ represent the probability of class $i$ given data $x_n$.

| Method | Standard | Random | PGD1 |
|---|---|---|---|
| Standard | $98.11\% \pm 0.07$ | $97.81\% \pm 0.06$ | $96.83\% \pm 0.11$ |
| ADV | $70.33\% \pm 0.45$ | $70.04\% \pm 0.45$ | $65.46\% \pm 0.40$ |
| **SOAR** | | | |
| - zero init | $99.99\% \pm 0.01$ | $\mathbf{99.97}\% \pm 0.01$ | $99.99\% \pm 0.01$ |
| - random init | $99.98\% \pm 0.01$ | $\mathbf{99.98}\% \pm 0.00$ | $100.0\% \pm 0.00$ |
| - PGD1 init | $97.71\% \pm 0.10$ | $97.63\% \pm 0.08$ | $97.94\% \pm 0.09$ |

We summarize the average value of the highest probability output for test set data initialized with zero, random and PGD1 perturbations in Table 5. We notice that training with SOAR using zero or random initialization leads to models with nearly 100% confidence on their predictions. This is aligned with the analysis of SOAR for a linear classifier (Section 4.1), which shows that the regularizer becomes ineffective as the model outputs high confidence predictions. Indeed, results in Table 7 show that those models are vulnerable under black-box attacks.

Results in Table 5 suggest that highly confident predictions *could be* an indication for gradient masking. We demonstrate this using the gradient-based PGD attack. Recall that we generate PGD attacks by first initializing the clean data $x_n$ with a randomly chosen $\eta$ within the $\ell_\infty$ ball of size $\varepsilon$, followed by gradient ascent at $x_n + \eta$. Suppose that the model makes predictions with 100% confidence on any given input. This leads to a piece-wise  loss surface that is either zero (correct predictions) or infinity (incorrect predictions). The gradient of this loss function is either zero or undefined, and thus making gradient ascent ineffective. Therefore, white-box gradient-based attacks are unable to find adversarial examples.

# E SUPPLEMENTARY EXPERIMENTS

## E.1 DISCUSSION ON THE REPRODUCIBILITY OF CURE AND LLR

We were not able to reproduce results of two closely related works, CURE (Moosavi-Dezfooli et al., 2019) and LLR (Qin et al., 2019). For CURE, we found the open-source implementation[2], but were not able to reproduce their reported results using their implmentation. We were not able to reproduce the results of CURE with our own implementation either. For LLR, Yang et al. (2020) were not able to reproduce the results, they also provided an open-source implementation[3]. Regardless, we compare SOAR to the **reported result** by CURE and LLR in Table 6:

Table 6: Comparison of SOAR, CURE and LLR on CIFAR-10 against $\ell_\infty$ bounded adversarial perturbations ($\varepsilon = 8/255$).

| Method | Standard Accuracy | PGD20 | Architecture |
|--------|-------------------|-------|--------------|
| **SOAR** | 87.95% | 56.06% | ResNet-10 |
| CURE | 81.20% | 36.30% | ResNet-18 |
| CURE | 83.10% | 41.40% | WideResNet |
| LLR | 86.83% | 54.24% | WideResNet |

## E.2 TRAINING AND EVALUATION SETUP

**CIFAR-10**: Training data is augmented with random crops and horizontal flips.

**ResNet**: We used an open-source ResNet-10 implementation[4]. More specifically, we initialize the model with ResNet(BasicBlock, [1,1,1,1]). Note that we remove the BatchNorm layers in the ResNet-10 architecture, and we discuss this further in Appendix E.7 .

**WideResNet**: We used the implementation[5] of WideResNet-34-10 model found in public repository maintained by the authors of TRADES (Zhang et al., 2019).

**Standard training on ResNet and WideResNet**: Both are trained for a total of 200 epochs, with an initial learning rate of 0.1. The learning rate decays by an order of magnitude at epoch 100 and 150. We used a minibatch size of 128 for testing and training. We used SGD optimizer with momentum of 0.9 and a weight decay of 2e-4.

**Adversarial training with PGD10 examples on ResNet**: The optimization setting is the same as the one used for standard training. Additionally, to ensure that the final model has the highest adversarial robustness, we save the model at the end of every epoch, and the final evaluation is based on the one with the highest PGD20 accuracy.

**SOAR on ResNet**: SOAR refers to continuing the training of the Standard model on ResNet. It is trained for a total of 200 epochs with an initial learning rate of 0.004 and decay by an order of magnitude at epoch 100. We used SGD optimizer with momentum of 0.9 and a weight decay of 2e-4. We use a FD step-size $h = 0.01$ for the regularizer. Additionally, we apply a clipping of 10 on the regularizer, and we discuss this clipping operation in Appendix E.7 .

**MART and TRADES on ResNet**: We used the same optimization setup as the ones in their respective public repository[6]. We briefly summarize it here. The model is trained for a total of 120 epochs, with an initial learning rate of 0.1. The learning rate decays by an order of magnitude at epoch 75, 90, 100. We used SGD optimizer with momentum of 0.9 and a weight decay of 2e-4. We performed a hyperparameter sweep on the strength of the regularization term $\beta$ and selected one that resulted in the best performance against PGD20 attacks. A complete result is reported in Appendix E.12 .

---

[2] https://github.com/F-Salehi/CURE_robustness
[3] https://github.com/yangarbiter/robust-local-lipschitz
[4] https://github.com/kuangliu/pytorch-cifar
[5] https://github.com/yaodongyu/TRADES
[6] https://github.com/YisenWang/MART

**MMA on ResNet**: We used the same optimization setup as the one in its public repository[7]. We briefly summarize it here. The model is trained for a total of 50000 iterations, with an initial learning rate of 0.3. The learning rate changes to 0.09 at the 20000 iteration, 0.03 at the 30000 iteration and lastly 0.009 at the 40000 iteration. We used SGD optimizer with momentum of 0.9 and a weight decay of 2e-4. We performed a hyperparameter sweep on the margin term and selected the one that resulted in the best performance against PGD20 attacks. A complete result is reported in Appendix E.12 .

**ADV, TRADES, MART and MMA on WideResNet**: We use the pretrained checkpoint provided in their respective repositories. Note that we use the pretrained checkpoint for PGD10 adversarially trained WideResNet in Madry's CIFAR10 Challenge[8].

**Evaluations**: For FGSM and PGD attacks, we use the implementation in AdverTorch (Ding et al., 2019c). For SimBA (Guo et al., 2019), we use the authors' open-source implementation[9].

### E.3 ADVERSARIAL ROBUSTNESS OF THE MODEL TRAINED USING SOAR WITH DIFFERENT INITIALIZATIONS

Table 7: Performance of SOAR with different initializations on CIFAR-10 against white-box and transfer-based black-box $\ell_\infty$ bounded adversarial perturbations ($\varepsilon = 8/255$).

| Method | Standard accuracy | White-box PGD20 | Black-box PGD20 |
|---|---|---|---|
| **SOAR** | | | |
| - zero init | 91.73% | 89.24% | 2.86% |
| - rand init | 91.70% | 90.82% | 9.16% |
| - PGD1 init | 87.95% | **56.06**% | **79.25**% |

We report the adversarial robustness of the model trained using SOAR with different initialization techniques in Table 7. The second column shows the accuracy against white-box PGD20 adversaries. The third column shows the accuracy against black-box PGD20 adversaries transferred from an independently initialized and standard-trained ResNet-10 model.
Note that despite the high adversarial accuracy against white-box PGD attacks, models trained using SOAR with zero and random initialization perform poorly against transferred attacks. This suggests the presence of gradient masking when using SOAR with zero and random initializations. Evidently, SOAR with PGD1 initialization alleviates the gradient masking problem.

### E.4 COMPARING THE VALUES OF THE SOAR REGULARIZED LOSS COMPUTED USING DIFFERENT NUMBERS OF RANDOMLY SAMPLED $z$

Table 8: Values of the regularized loss computed using different numbers of $z$ at the beginning and the end of SOAR regularization.

| Checkpoints | $n = 1$ | $n = 10$ | $n = 100$ |
|---|---|---|---|
| Beginning of SOAR | 10.58 | 10.58 | 10.58 |
| End of SOAR | 1.57 | 1.56 | 1.56 |

Suppose we slightly modify Eq (13) by $\ell_{\text{SOAR}}(x, y, n) = \ell(x', y) + \frac{1}{n} \sum_{i=0}^{n} R(x'; z_{(i)}, h)$ to incorporate the effect of using multiple randomly sampled vectors $z_{(i)}$ in computing the SOAR regularized loss. Notice that the current implementation is equivalent to using $n = 1$. We observed the model at two checkpoints, at the beginning and the end of SOAR regularization, the value of the regularized loss remains unchanged as we increase $n$ from 1 to 100.

---

[7] https://github.com/BorealisAI/mma_training
[8] https://github.com/MadryLab/cifar10_challenge
[9] https://github.com/cg563/simple-blackbox-attack

## E.5 ROBUSTNESS UNDER $\ell_2$ ATTACKS ON CIFAR-10

We evaluate SOAR and two of the baseline methods, ADV and TRADES, against $\ell_2$ white-box and black-box attacks on CIFAR-10 in Table 9. No $\ell_2$ results were reported by MART and we are not able to reproduce the $\ell_2$ results using the implementation by MMA, thus those two methods are not included in our evaluation.

In Section 4, we show that the $\ell_\infty$ formulation of SOAR with $\|\delta\|_\infty = \varepsilon$ is **equivalent** to the $\ell_2$ formulation of SOAR with $\|\delta\|_2 = \varepsilon\sqrt{d}$. In other words, models trained with SOAR to be robust against $\ell_\infty$ attacks with $\varepsilon = \frac{8}{255}$ should also obtain improved robustness against $\ell_2$ attacks with $\varepsilon = \frac{8}{255}\sqrt{32 * 32 * 3} = 1.74$. In our evaluation, all $\ell_2$ adversaries used during ADV and TRADES are generated with 10-step PGD ($\varepsilon = 1.74$) and a step size of $0.44$. Note that the goal here is to show the improved robustness of SOAR against $\ell_2$ attacks other than being SOTA, thus the optimization procedures are the same as the ones used in the $\ell_\infty$ evaluation.

We observe that training with SOAR improves the robustness of the model against $\ell_2$ attacks. Instead of a fixed $\ell_2$ norm, we demonstrate the improved robustness using an increasing range of $\varepsilon$. For all attacks, we use 100 iterations of PGD and a step size of $\frac{2.5\varepsilon}{100}$. In Table 9, we find that training with SOAR leads to a significant increase in robustness against white-box and black-box $\ell_2$ adversaries. As $\varepsilon$ increases, SOAR model remain robust against white-box $\ell_2$ attacks ($\varepsilon = 1$), while other methods falls off. The last column of Table 9 shows the robustness against transferred $\ell_2$ attacks ($\varepsilon = 1.74$). The source model is a ResNet10 network trained separately from the defence models on the unperturbed training set. We observe that SOAR achieves the second highest robustness compared to baseline methods against transferred $\ell_2$ attacks. This result empirically verifies our previous claim that $\ell_2$ and $\ell_\infty$ formulation of SOAR only differs by a factor of $\sqrt{d}$. Moreover, it aligns with findings by Simon-Gabriel et al. (2019), that empirically showed adversarial robustness through regularization gains robustness against more than one norm-ball attack at the same time.

Table 9: $\ell_2$ robustness of the adversarially trained model (under $\ell_2$ formulations) at different epsilon values. 100-step PGD is used for all attacks. Accuracy (%) against $\ell_2$-PGD attacks.

|  | Method | $\varepsilon = \frac{60}{255}$ | $\varepsilon = \frac{120}{255}$ | $\varepsilon = \frac{255}{255}$ | $\varepsilon = 1.74$ | Transfer |
|---|---|---|---|---|---|---|
| ResNet | ADV | 68.13% | 62.03% | 47.53% | 28.09% | **70.86**% |
| | TRADES | 68.59% | 62.39% | 45.42% | 25.67% | 69.02% |
| | SOAR | **75.39**% | **66.81**% | **60.90**% | **56.89**% | 69.52% |

Table 10: Performance of the ResNet models on CIFAR-10 against the four $\ell_2$-bounded attacks used as an ensemble in AutoAttack ($\varepsilon = 1.74, 1.0, 0.5$).

| Method | Untargeted APGD-CE | Targeted APGD-DLR | Targeted FAB | Square Attack |
|---|---|---|---|---|
| ADV | 24.61\|45.90\|60.62 | 22.00\|42.99\|58.78 | 22.57\|43.31\|58.89 | 45.69\|57.84\|65.81 |
| TRADES | 20.22\|43.50\|61.51 | 15.05\|37.97\|58.78 | 14.44\|36.37\|58.19 | 43.60\|57.98\|68.17 |
| SOAR | 51.37\|56.51\|64.18 | 0.97\|16.55\|52.97 | 1.05\|17.70\|53.42 | 21.34\|49.64\|71.92 |

## E.6 ADDITIONAL EVALUATION ON SVHN DATASET

We use the same ResNet-10 architecture as the one for CIFAR-10 evaluation. Training data is augmented with random crops and horizontal flips. For Standard training, we use the same optimization procedure as the one used for CIFAR-10. For SOAR and TRADES, we use the exact same hyperparameter for the regularizer. For SOAR, we use early-stopping at epoch 130 to prevent catastrophic over-fitting. Besides, the optimization schedule is identical for SOAR and TRADES as the ones used for CIFAR-10.

We emphasize again that the goal of evaluting using SVHN is to demonstrate the improved robustness with SOAR on a different dataset, thus we did not perform an additional hyper-parameter sweep. The optimization procedures are the same as the ones used in the CIFAR-10 evaluation.

Table 11: Performance on SVHN against $\ell_\infty$ bounded white-box attacks ($\varepsilon = 8/255$).

| Method | Standard | FGSM | PGD20 | PGD100 | PGD200 | PGD1000 | PGD20-50 |
|---|---|---|---|---|---|---|---|
| Standard | **94.93**% | 34.41% | 2.71% | 2.27% | 2.20% | 2.09% | 2.15% |
| ADV | 91.8% | 61.0% | 43.2% | 42.1% | — | — | — |
| TRADES | 84.48% | 58.01% | 48.90% | 48.15% | 48.11% | 48.05% | 48.01% |
| SOAR | 91.23% | **72.70**% | **58.99**% | **56.80**% | **56.56**% | **56.38**% | **56.70**% |

Table 12: Performance on SVHN against $\ell_\infty$ bounded black-box attacks ($\varepsilon = 8/255$).

| Method | SimBA | PGD20 | PGD1000 |
|---|---|---|---|
| TRADES | 49.13% | 75.35% | 75.27% |
| SOAR | **64.20**% | **75.42**% | **75.81**% |

For PGD10 adversarial training, we observe that ResNet-10 is not able to learn anything meaningful. Specifically, when trained with PGD10 examples, ResNet-10 does not perform better than a randomly-initialized network in both standard and adversarial accuracy. Cai et al. (2018) made a similar observation on ResNet-50, where training accuracy is not improving over a long period of adversarial training with PGD10. They further investigated models with different capacities and found that even ResNet-50 might not be sufficiently deep for PGD10 adversarial training on SVHN. Wang & Zhang (2019) reported PGD10 adversarial training result on SVHN with WideResNet, which we include in Table 11.

For MART, we were not able to translate their CIFAR-10 results on SVHN. We performed the same hyperparameter sweep as the one in Table 18, as well as different optimization settings, but none resulted in a meaningful model. It is likely that the potential cause is the small capacity of ResNet-10. For MMA, the implementation included in its public repository is very specific to the CIFAR-10 dataset, so we did not include it in the comparison.

Overall, we observe a similar performance on SVHN vs. on CIFAR-10. Compared to the result in Table 1, we observe a slight increase in standard accuracy and robust accuracy for both SOAR and TRADES. In particular, the standard accuracy increases by $8.87\%$ and $3.28\%$, and the PGD20 accuracy increases by $3.52\%$ and $2.93\%$ for TRADES and SOAR respectively. More notably, we observe on SVHN that SOAR regularized model gains robustness without significantly sacrificing its standard accuracy.

Table 12 compares the performance of SOAR to TRADES on SimBa and on transferred $\ell_\infty$ attacks. The evaluation setting for transferred attacks is identical to the one used for CIFAR-10, where we use an undefended independently trained ResNet-10 as the source model. Despite a smaller gap on the accuracy against transferred attacks, we see that SOAR regularized model yields a significant higher accuracy against the stronger SimBA attacks.

Note that we did not perform any extensive hyperparameter sweep on SVHN, and we simply took what worked on CIFAR-10. We stress that the goal is to demonstrate the effectiveness of SOAR, and its performance relative to other baseline methods.

Next, we evaluate SOAR and TRADES under $\ell_2$ bounded white-box and black-box attacks. All $\ell_2$ PGD adversaries are generated using the same method as the one in the evaluation for CIFAR-10. Also, we do not include ADV due to the same result discussed above. Our results show that training with SOAR significantly improves the robustness against $\ell_2$ PGD white-box attacks compared to TRADES. For transferred attacks, TRADES and SOAR performs similarly.

### E.7 CHALLENGES

**Batch Normalization**: We observe that networks with BatchNorm layers do not benefit from SOAR in adversarial robustness. Specifically, we performed an extensive hyper-parameter search for SOAR on networks with BatchNorm layers, and we were not able to achieve meaningful improvement

Table 13: Performance on SVHN against $\ell_2$ bounded white-box and black-box attacks. 100-step PGD is used for all attacks.

| Method | Standard | $\varepsilon = \frac{60}{255}$ | $\varepsilon = \frac{120}{255}$ | $\varepsilon = \frac{255}{255}$ | $\varepsilon = 1.739$ | Transfer |
|---|---|---|---|---|---|---|
| Standard | **94.93**% | 53.46% | 30.04% | 18.20% | 14.60% | 15.52% |
| TRADES | 91.94% | 55.17% | 47.47% | 32.34% | 17.52% | **50.22**% |
| SOAR | 91.23% | **82.35**% | **69.65**% | **56.70**% | **48.38**% | 49.13% |

in adversarial robustness. A related work by Galloway et al. (2019) focuses on the connection between BatchNorm and adversarial robustness. In particular, their results show that on VGG-based architecture (Simonyan & Zisserman, 2014), there is a significant gap in adversarial robustness between networks with and without BatchNorm layers under standard training. Needless to say, the interaction between SOAR and BatchNorm requires further investigations, and we consider this as an important future direction. As such, we use a small-capacity ResNet (ResNet-10) in our experiment, and modified it by removing its BatchNorm layers. Specifically, we removed BatchNorm layers from all models used in the baseline experiments with ResNet. Note that BatchNorm layers makes the training process less sensitive to hyperparameters (Ioffe & Szegedy, 2015), and removing them makes it difficult to train a very deep network such as WideResNet. As such, we did not perform SOAR on WideResNet.

**Starting from pretrained model**: We notice that it is difficult to train with SOAR on a newly-initialized model. Note that it is a common technique to perform fine-tuning on a pretrained model for a specific task. In CURE, regularization is performed after a model is first trained with a cross-entropy loss to reach a high accuracy on clean data. They call the process *adversarial fine-tuning*. Cai et al. (2018); Sitawarin et al. (2020) study the connection between curriculum learning (Bengio et al., 2009) and training using adversarial examples with increasing difficulties. Our idea is similar. The model is first optimized for an easier task (standard training), and then regularized for a related, but more difficult task (improving adversarial robustness). Since the model has been trained to minimize its standard loss, the loss gradient can be very small compared to the regularizer gradient, and thus we apply a clipping of 10 on the regularizer.

**Catastrophic Overfitting**: We observe that when the model achieves a high adversarial accuracy and continues training for a long period of time, both the standard and adversarial accuracy drop significantly. A similar phenomenon was observed in (Cai et al., 2018; Wong et al., 2019a), which they refer to as *catastrophic forgetting* and *catastrophic over-fitting* respectively. Wong et al. (2019a) use early-stopping as a simple solution. We observe that with a large learning rate, the model reaches a high adversarial accuracy faster and catastrophic over-fitting happens sooner. As such, our solution is to fix the number of epochs to 200 and then carefully sweep over various learning rates to make sure that catastrophic over-fitting do not happen.

**Discussion on Computation Complexity**: We emphasize that our primary goal is to propose regularization as an alternative approach to improving adversarial robustness. We discussed techniques towards an efficient implementation, however, there is still potential for a faster implementation. In our current implementation, a single epoch with WideResNet takes: 19 mins on PGD10 adversarial training, 26.5 mins on SOAR, 29 mins on MART, and 39.6 mins on TRADES. We observe that despite being a faster method than MART and TRADES, SOAR is still quite slow compared to PGD10 adversarial training. We characterize the computation complexity as a function of the number of forward and backward passes required for a single mini-batch. Standard training: 1 forward pass and 1 backward pass; Adversarial training with k-step PGD: k+1 forward passes and k+1 backward passes; FOAR: 1 forward pass and 2 backward passes; SOAR: 3 forward passes and 4 backward passes

## E.8 DIFFERENTIABILITY OF RELU AND ITS EFFECT ON SOAR

The SOAR regularizer is derived based on the second-order Taylor approximation of the loss which requires the loss to be twice-differentiable. Although ReLU is not differentiable at 0, the probability of its input being at exactly 0 is very small. That is also why we can train ReLU networks through backpropagation. This is true for the Hessian too. In addition, notice that from a computation

viewpoint, we never need to compute the exact Hessian as we approximate it through first-order approximation.

### E.9 POTENTIAL ROBUSTNESS GAIN WITH INCREASING CAPACITIES

Empirical studies in Madry et al. (2018) and Wang et al. (2020) reveal that their approaches benefit from increasing model capacity to achieve higher adversarial robustness. We have a similar observation with SOAR.

Table 14: Performance on CIFAR-10 against $\ell_\infty$ bounded white-box attacks ($\varepsilon = 8/255$).

| Model | Standard | PGD20 | PGD100 | PGD200 |
|---|---|---|---|---|
| CNN6 | 81.73% | 32.83% | 31.20% | 31.15% |
| CNN8 | 83.65% | 47.30% | 46.07% | 45.83% |
| ResNet-10 | 87.95% | 56.06% | 55.00% | 54.94% |

Table 14 compares the performance of SOAR against $\ell_\infty$ bounded white-box attacks on networks with different capacities. CNN6(CNN8) refers to a simple 6-layer(8-layer) convolutional network, and ResNet-10 is the network we use in Section 5. Evidently, as network capacity increases, we observe improvements in both standard accuracy and adversarial accuracy. As such, we expect a similar gain in performance with larger capacity networks such as WideResNet.

### E.10 EXPERIMENT RESULTS ON RESNET10 IN TABLE 1 AND TABLE 2 WITH STANDARD DEVIAITIONS

All results on ResNet10 are obtained by averaging over 3 independently initialized and trained models. Here, we report the standard deviation of the results in Table 1 and Table 2. Notice we omit results on PGD100 and PGD200 due to space constraint.

Table 15: Performance on CIFAR-10 against $\ell_\infty$ bounded white-box attacks ($\varepsilon = 8/255$). (Table 1 with standard deviations)

| | Method | Standard | FGSM | PGD20 | PGD1000 | PGD20-50 |
|---|---|---|---|---|---|---|
| | Standard | 92.54%(0.07) | 21.59%(0.29) | 0.14%(0.03) | 0.08%(0.02) | 0.10%(0.02) |
| | ADV | 80.64%(0.40) | 50.96%(0.27) | 42.86%(0.18) | 42.17%(0.15) | 42.55%(0.15) |
| | TRADES | 75.61%(0.19) | 50.06%(0.09) | 45.38%(0.15) | 45.16%(0.17) | 45.24%(0.16) |
| ResNet | MART | 75.88%(1.12) | 52.55%(0.28) | 46.60%(0.43) | 46.21%(0.47) | 46.40%(0.48) |
| | MMA | 82.37%(0.15) | 47.08%(0.37) | 37.26%(0.38) | 36.64%(0.41) | 36.85%(0.44) |
| | FOAR | 65.84%(0.24) | 36.96%(0.17) | 32.28%(0.11) | 31.89%(0.10) | 32.08%(0.07) |
| | SOAR | 87.95%(0.08) | 67.15%(0.32) | 56.06%(0.15) | 54.69%(0.17) | 54.20%(0.16) |

Table 16: Performance on CIFAR-10 against $\ell_\infty$ bounded black-box attacks ($\varepsilon = 8/255$). (Table 2 with standard deviations)

| | Method | SimBA | PGD20-R | PGD20-W | PGD1000-R | PGD1000-W |
|---|---|---|---|---|---|---|
| | ADV | 47.27%(1.01) | 77.19%(0.30) | 79.48%(0.38) | 77.22%(0.34) | 79.55%(0.37) |
| | TRADES | 47.67%(0.52) | 72.28%(0.19) | 74.39%(0.28) | 72.24%(0.14) | 74.37%(0.21) |
| ResNet | MART | 48.57%(0.99) | 72.99%(0.90) | 74.91%(1.15) | 72.99%(0.87) | 75.04%(1.12) |
| | MMA | 43.53%(1.25) | 78.70%(0.09) | 80.39%(1.20) | 78.72%(0.09) | 81.35%(0.17) |
| | FOAR | 35.97%(0.26) | 63.56%(0.30) | 65.20%(0.21) | 63.60%(0.33) | 65.27%(0.32) |
| | SOAR | 68.57%(0.95) | 79.25%(0.45) | 86.35%(0.04) | 79.49%(0.29) | 86.47%(0.10) |

### E.11  ADDITIONAL EXPERIMENTS ON GRADIENT MASKING

To verify that SOAR improves robustness of the model without gradient masking, we include the following experiments to empirically support our claim.

First, from the result in Appendix E.3, we conclude that SOAR with zero initilaization results in gradient masking. This is shown by the high accuracy (89.24%, close to standard accuracy) under white-box PGD attacks and low accuracy (2.86%) under black-box transferred attacks. Next, prior work has verified that adversarial training with PGD20 adversaries (ADV) results model without gradient masking Athalye et al. (2018). Therefore, let us use models trained using ADV and SOAR(zero-init) as examples of models with/without gradient masking respectively.

In the $\ell_\infty$ attack setting, PGD uses the sign of the gradient $\text{sign}(\nabla_x \ell(x))$ to generate perturbations. As such, one way to verify the strength of gradient is to measure the average number of none-zero elements in the gradient. A model with gradient masking is expected to have much less non-zero elements than one without. In our experiment, the average non-zero element in gradient is 3072 for ADV trained (no GM), 3069 for SOAR (PGD1-init) and 1043 for SOAR (zero-init, has GM). We observe that SOAR with PGD1-init has a similar number of non-zero gradient elements compared to ADV, meaning PGD adversary can use sign of those non-zero gradient elements to generate meaningful perturbations.

In Section 5, the 20-iteration $\ell_\infty$ PGD adversaries are generated with a step-size of $\frac{2}{255}$ and $\varepsilon = \frac{8}{255}$. Suppose we use $\varepsilon = 1$ instead of $\varepsilon = \frac{8}{255}$ and other parameters remain the same, that is, we allow the maximum $\ell_\infty$ perturbation to reach the input range ($[0, 1]$) and generate PGD20 attacks. We observe such attacks result in near black-and-white images on SOAR with PGD-1 init; it has a 0% accuracy against such PGD20 attacks, similar to the 3.3% on ADV trained model. On the other hand, the robust accuracy for SOAR (zero-init) is 9.7%.

### E.12  HYPERPARAMETER SWEEP FOR TRADES, MART AND MMA ON RESNET

The following results show the hyperparameter sweep on TRADES, MART and MMA respectively. We include the one with the highest PGD20 accuracy in Section 5.

Table 17: Hyperparameter sweep of TRADES on ResNet: evaluation based performance on CIFAR-10 against $\ell_\infty$ bounded adversarial perturbations ($\varepsilon = 8/255$).

| $\beta$ | Standard | FGSM | PGD20 | PGD100 | PGD200 | PGD20-50 |
|---|---|---|---|---|---|---|
| 15 | 72.80% | 48.85% | 44.80% | 44.63% | 44.63% | 44.67% |
| 13 | 73.46% | 49.18% | 45.09% | 44.97% | 44.96% | 44.98% |
| 11 | 74.46% | 49.39% | 44.86% | 44.74% | 44.71% | 44.75% |
| 9 | 75.61% | **50.06%** | **45.38%** | **45.19%** | **45.18%** | **45.24%** |
| 7 | 76.74% | 50.05% | 45.04% | 44.80% | 44.78% | 44.87% |
| 5 | 78.39% | 50.22% | 44.40% | 44.11% | 44.14% | 44.26% |
| 3 | 80.20% | 49.84% | 42.53% | 42.12% | 42.09% | 42.23% |
| 1 | 83.69% | 45.61% | 35.27% | 34.62% | 34.52% | 34.87% |
| 0.5 | 84.00% | 46.16% | 34.40% | 33.48% | 33.38% | 33.75% |
| 0.1 | 84.64% | 41.12% | 16.52% | 14.80% | 14.53% | 14.83% |

Table 18: Hyperparameter sweep of MART on ResNet: evaluation based performance on CIFAR-10 against $\ell_\infty$ bounded adversarial perturbations ($\varepsilon = 8/255$).

| $\beta$ | Standard | FGSM | PGD20 | PGD100 | PGD200 | PGD20-50 |
|---|---|---|---|---|---|---|
| 15 | 72.78% | 52.13% | 46.41% | 46.05% | 45.98% | 46.14% |
| 13 | 73.03% | 51.43% | 46.29% | 46.05% | 45.99% | 46.15% |
| 11 | 74.25% | 51.57% | 46.17% | 45.84% | 45.83% | 45.95% |
| 9 | 75.88% | **52.55**% | **46.60**% | **46.29**% | **46.25**% | **46.40**% |
| 7 | 75.77% | 52.54% | 46.31% | 45.94% | 45.92% | 46.07% |
| 5 | 76.84% | 52.07% | 46.00% | 45.67% | 45.59% | 45.80% |
| 3 | 78.25% | 52.22% | 45.32% | 44.84% | 44.82% | 45.10% |
| 1 | 80.30% | 52.12% | 44.10% | 43.58% | 43.52% | 43.86% |
| 0.5 | 80.79% | 52.06% | 43.95% | 43.32% | 43.24% | 43.61% |
| 0.1 | 80.82% | 51.97% | 43.39% | 42.78% | 42.76% | 43.01% |

Table 19: Hyperparameter sweep of MMA on ResNet: evaluation based performance on CIFAR-10 against $\ell_\infty$ bounded adversarial perturbations ($\varepsilon = 8/255$).

| Margin | Standard | FGSM | PGD20 | PGD100 | PGD200 | PGD20-50 |
|---|---|---|---|---|---|---|
| 12 | 84.54% | 46.11% | 34.56% | 33.89% | 33.81% | 34.12% |
| 20 | 83.12% | **47.48**% | 36.99% | 36.41% | 36.37% | 36.59% |
| 32 | 82.37% | 47.08% | **37.26**% | **36.71**% | **36.66**% | **36.85**% |
| 48 | 82.03% | 46.77% | 36.64% | 36.02% | 35.98% | 36.22% |
| 60 | 81.67% | 46.79% | 36.91% | 36.35% | 36.32% | 36.51% |
| 72 | 81.73% | 46.40% | 36.55% | 36.01% | 35.93% | 36.21% |

### E.13 ADVERSARIAL ROBUSTNESS OF THE MODEL TRAINED USING FOAR WITH DIFFERENT INITIALIZATIONS

FOAR achieves the best adversarial robustness using PGD1 initialization, so we only present this variation of FOAR in Section 5.

Table 20: Performance of FOAR with different initializations on CIFAR-10 against white-box and transfer-based black-box $\ell_\infty$ bounded adversarial perturbations ($\varepsilon = 8/255$).

| Initialization | Standard | FGSM | PGD20 | PGD100 | PGD200 | PGD1000 | PGD20-50 |
|---|---|---|---|---|---|---|---|
| zero | 74.43% | 33.39% | 23.65% | 22.83% | 22.79% | 23.23% | 68.58% |
| rand | 73.96% | 33.62% | 24.71% | 23.96% | 23.93% | 24.37% | **69.04**% |
| PGD1 | 65.84% | **36.96**% | **32.28**% | **31.87**% | **31.89**% | **32.08**% | 63.56% |

