# OpenReview forum: "SOAR: Second-Order Adversarial Regularization"
_ICLR.cc/2021/Conference — Reject_

### Official Review · AnonReviewer3 · 2020-10-28
**Official Blind Review #3**

**Rating:** 7
**Confidence:** 3

**Review:**

Summary:

The paper proposed a regularizer loss as an alternative to adversarial training to improve the robustness of neural networks against adversarial attacks. The new regularizer is derived from a second-order Tyler series expansion of the loss function in the model robustness optimization problem. Clear mathematical derivation and thoughtful empirical experimental results are provided. The proposed method outperformed baseline adversarial training methods with better or on part robustness and higher standard accuracy.

Pros:
- The paper is really well-written and easy to understand. Both the intuition behind the method is conveyed clearly and the potential drawback of the method is discussed with solutions.
- The performance of the proposed method is outstanding with a large margin compared with other baseline methods.The experiments are extensive and rigorous.
- The idea is easy to implement.
- There is a lot of valuable discussion and experiments presented in the supplemental materials. In fact, some might be better to be moved to the main text.

Cons:
- As also discussed in the challenges in Appendix E6, an expected advance of this approach would be the training efficiency. However it is not well discussed in the main text. It would make the paper an even better one with some efficiency optimization.
- I have some concerns about the sampling steps in the regularization loss. Only a single sample is drawn for both eta and z. How does this sample affect the stability of the training? Some empirical analysis on the intermediate values would be nice.
- The author mentioned in the appendix that batch normalization layers are removed from the SOAR experiments. Do you also remove these layers from the baseline experiments? If not, how does the removal of BN layers impact the performance?

Other detail comments:
- I think the clipping of the regularization loss should also be mentioned and discussed in the main text.
- Missing space in conclusion: l2 attacks

---------------------------------------------
post-rebuttal
I would like to thank the reviewers for their efforts to improve the draft. Most of my concerns were resolved. However, I agree with other reviewers on the fact that the proposed method only present advantages in certain limited networks and scenarios.  To calibrate this weakness, I downgrade my score to 7. Overall, I think the paper explored an interesting and promising direction to improve network robustness using second-order regularization and solid progress was made. I will recommend accepting the paper for publication.

---

> ### Author Response · Authors · 2020-11-20
> **Thank you for your review**
>
> Thank you for the careful reading of the paper and your helpful comments. We indeed removed BatchNorm layers from all models used in the baseline experiments with ResNet. We will move the discussion of clipping to the main text, and include a brief discussion on potential ways to further optimize the method.
>
> Suppose we slightly modify Eq (13) by $\ell_{SOAR}(x, y, n) = \ell(x', y) + \frac{1}{n}\sum_{i=0}^{n} R(x';z_{(i)}, h)$ to incorporate the effect of using multiple randomly sampled vectors $z_{(i)}$  in computing the SOAR regularized loss. Notice that the current implementation is equivalent to using $n=1$. We observed the model at two checkpoints, at the beginning and the end of SOAR regularization, the value of the regularized loss does not vary as we increase $n$ from 1 to 100.
>
> Checkpoints              | n = 1             | n = 10             | n = 100
>
> Beginning of SOAR    | 10.58             | 10.59              | 10.58
>
> End of SOAR             | 1.57               | 1.56                 | 1.56
>
> *********************************
> Update:
>
> Due to space constrain caused by an additional section on AutoAttack (Section 5.3), we were not able to include a detailed discussion on clipping of the regularized loss. Instead, we added the specific point at the end of the 3rd paragraph in Section 5.
>
> An empirical study on the effect of sample size is added in Appendix E.4

---

### Official Review · AnonReviewer4 · 2020-10-28
**New regularization based training against adversarial examples**

**Rating:** 7
**Confidence:** 4

**Review:**

Authors propose a second order approximation based training procedure to build robust networks against \ell_inf and \ell_2 attacks.

The paper is very well written and ideas are clearly stated. The core intuition rests on building an approach similar to adversarial training, but without the computational expense and empirically inexact inner maximization procedure. The inexactness of inner maximization in adversarial training is indeed a crucial issue, which the authors highlight.

The authors first formulate a second order objective which uses the Hessian information of the loss function Eq. 6. Through Proposition 1 and the approximation in Eq. 10, they derive a bound on this regularizer which only depends on first order terms, hence making it computationally inexpensive (Algorithm 1 SOAR). To my knowledge, the derivations are clean.

The experiments are only done on ResNet and not on WideResNet, although authors have reported the performance of other robust algorithms on WideResNet. The show that robust accuracy of ResNet-10 with SOAR is better than all competing algorithms using ResNet-10 and comparable to that other algorithms when used on WideResNet. It will be interesting to see the gains of SOAR on WideResNet as well.

Another key observation from the experiments is that SOAR does not lead to as significant drops in standard accuracy (i.e. not under attack), which I do think is also another important contribution of this paper.

I do have a small issue with authors portraying regularization and robust training to be two separate objectives. Regularization is not specific to generalization properties, so I don't see why they dedicate several paragraphs to show that the two are the same.

Typos:

l_FOAR Eq.4 - should be <= not =

---

> ### Author Response · Authors · 2020-11-20
> **Thanks for your review**
>
> Thank you for the careful reading of the paper and your helpful comments.  As discussed in Appendix E.6, we observe that networks with BatchNorm layers do not benefit from SOAR in adversarial robustness. BatchNorm plays a significant part in training large networks such as WideResnet, as it makes the optimization procedure less sensitive to the choice of hyper-parameters. For this reason, we were not able to train SOAR on WideResNet. We believe resolving the BatchNorm issue is an important next step for SOAR to be applied on larger networks.
>
> We do not portray regularization and adversarial training as two separate objectives. In fact, they are two realizations of the inner-max objective in the robust optimization framework as discussed in Section 3. Adversarial training approximates the solution to the inner-max by using a maximizer computed using a particular algorithm, while training with a regularizer directly optimizes the weight with the effect of the maximizer in mind. In the motivation example in Section 2, we showed that the two can be equivalent, and this motivated the derivation of SOAR, a regularizer that mimics the effect of adversarial training.
>
> For l_FOAR Eq.4, we can first use Cauchy-Schwarze to show that $\max_{||\delta||_\infty \leq \epsilon} \nabla \ell(x)^{\top} \delta \leq \epsilon ||\nabla \ell(x)||_1$, and taking $\delta = \epsilon \text{sign} (\nabla \ell(x))$ shows that the maximum is precisely $\epsilon ||\nabla \ell(x)||_1$.
>
> *********************************
> Update:
>
> We modified the end of Section 3 to clarify our position on regularization and adversarial training.

---

### Official Review · AnonReviewer5 · 2020-11-05
**Some Concerns Needed to be Addressed**

**Rating:** 4
**Confidence:** 5

**Review:**

This paper introduced a regularization scheme based on the second-order Taylor expansion of the loss objective to improve the robustness of the trained models.

Theoretically, it used the simple linear model as an example to demonstrate 1) with a proper regularization term, the training algorithm can encode robustness without explicitly solving the inner maximization problem, i.e., generating adversarial examples. 2) using the second-order estimation at the origin may lead to gradient masking and using a perturbed input can tackle this problem.

The algorithm proposed is based on an upper bound of a quadratic.

The experiments show some results on CIFAR10 and SVHN in comparison with some baselines in both white-box and black-box settings.

Comments on the intuition and methodology:

1. We know that most modern deep learning models are based on ReLU activations (or its variants like leaky ReLU), which is not second-order differentiable. In these cases, the Hessian matrix of the loss is ill defined. The author should discuss this.

2. I cannot see the "uniqueness" or the advantage of regularization introduced in Equation (1). Actually, the quadratic form of (1) comes from the mean squared loss of $l$ instead of second-order Taylor expansion. In addition, the regularization term in (1) depends on $\Delta x$. That's to say, using a different perturbation $\Delta x$, the regularization term would be different. It is not necessarily in a quadratic form and can also drive the trainable parameter $w$ to the most robust one.

3. The upper bound derived in Section 4 is not a strictly upper bound of the loss function, because it is based on a second-order approximation. The highlight of the word "worst-case" is a bit misleading. In addition, the author did not point out why we need a "upper bound" here? In adversarial training, we are actually optimizing a lower bound of the inner maximization problem. It is reasonable as long as this lower bound is tight, corresponding to a stronger attacker. I think the key point is not the upper bound of the approximation, but how close the objective function you are training on is to the inner maxima of Equation (2).

4. In Equation (7), replacing $\|\delta\|_\infty \leq \epsilon$ with $\|\delta\|_2 \leq \epsilon \sqrt{d}$ is risky even if $\delta$ is a dense vector. The volume of $\\{\delta | \|\delta\|_\infty \leq \epsilon\\}$ is $2^d\epsilon^d$, while the volume of $\\{\delta | \|\delta\|_2 \leq \epsilon \sqrt{d}\\}$ is $\frac{\pi^{d / 2}}{\Gamma(1 + d / 2)} d^{d / 2}\epsilon^d$. The volume ratio of them goes quickly to zero as the dimension $d$ increases. As a result, the RHS of (7) is a much looser bound than LHS.

5. The authors should discuss the complexity of Algorithm 1 and compare it with baselines.

6. The authors should justify the claim "In such a case, the Taylor series expansion, computed using the gradient and Hessian evaluated at x, becomes an inaccurate approximation to the loss, and hence its maximizer is not a good solution to the inner maximization problem." in the second last paragraph of page 5. A regularizer of small values does not mean it is not a good regularizer in this case.

7. The sentence "our approximation might not be very good whenever
the loss function is close to being flat at x" in the end of page 5 is incorrect. The approximation error is approximately the third-order Taylor expansion term and does not connect to the flatness at $x$.

Comments on the experiments:

1. The author should evaluate the robustness of the model by the state-of-the-art attack, such as AutoAttack introduced in https://github.com/fra31/auto-attack.

2. The results on the baselines is not convincing. The accuracy is a bit worse than the one claimed in their papers. For example, regarding adversarial training, the authors should use early stopping introduced in https://arxiv.org/abs/2002.11569 as it typically overfits in the end.

The writing is easy to follow. As far as what I know, there is no key missing references.

Based on the concerns listed above, my score is a reject unfortunately, but I look forward to the discussion with the authors, I will re-evaluate the paper after the rebuttal period.

---

> ### Author Response · Authors · 2020-11-20
> **Response I**
>
> Thank you for the careful reading of the paper and your helpful comments.
>
> Response to comment 1:
>
> The SOAR regularizer is derived based on the second-order Taylor approximation of the loss which requires the loss to be twice-differentiable. Although ReLU is not differentiable at 0, the probability of its input being at exactly 0 is very small. That is also why we can train ReLU networks through backpropagation. This is true for the Hessian too. In addition, notice that from a computation viewpoint, we never need to compute the exact Hessian as we approximate it through first-order approximation.
>
> Response to comment 2:
>
> We emphasize that the goal of the motivation example is to formally demonstrate regularization as an alternative approach -- rather than showing its uniqueness or advantage -- to adversarial training. We agree with the reviewer on the limitation of this example, as we clearly mentioned at the end of Section 2. These limitations were indeed the motivation for the rest of the paper.
> This example is inspired by Bishop et al. [1995], which showed that training with noisy data is equivalent to Tikhonov Regularization. Similarly, under the problem setup, we showed that the regularizer in (1) provides a direct alternative to training with perturbed inputs, where both methods cause weights of irrelevant dimensions to go to zero and result in a robust model.
> Equation (1) can be derived in two ways: 1. by substituting $\Delta x$ into the loss (Appendix A), 2. by approximating the loss using a second-order Taylor approximation with the same direction and step size (Appendix A.1). For the latter, we showed that a linear approximation would only recover the first regularizer term, which motivated the use of a second-order approach for an improved approximation.
>
> Response to comment 3:
>
> The reviewer is correct that (9) is an upper-bound on the loss under its second-order approximation. In the paper, the term “worst-case” loss is always followed by specifying that loss is under an approximation.
> The reason for upper bounding the inner max is somewhat, but not exactly, similar to the reason we use convex relaxation of binary 0-1 loss: it leads to a computationally easier optimization problem. By (approximately) upper bounding the inner-max $\max_{\delta} f(x+\delta, w)$ by $\phi(x,w)$, we can then solve $\min_{w} \phi(x,w)$, which is presumably easier than $\min_{w} \max_{\delta} f(x+\delta, w)$. For the first-order approximation, we get an analytical solution, and for the second-order approximation, we get a numerically easy procedure to compute it.
> As $\min_{w} \max_{\delta} f(x+\delta, w) \leq \min_{w} \phi(x,w)$, this means that we are minimizing the worst-case damage adversaries can cause which leads to improving the robustness of the model (modulus the second-order Taylor series approximations, which might not provide a true upper bound).
> It is true that not all upper bounds are of the same quality, the same way that not all surrogate losses are of the same quality. Throughout the paper, we have tried to go through a series of upper bounding steps that are reasonable.
>
> Response to comment 4:
>
> This is indeed an interesting observation and we will continue investigating it. As a preliminary study on the tightness of the bound, we evaluated the two slides of (7) by approximating the maximum using PGD attacks. In particular, we approximate $\max_{||\delta||_\infty \leq \epsilon} \ell(x+\delta)$ using $\ell(x+\delta_\infty)$ where $\delta_\infty$ is generated using 20-iteration $\ell_\infty$-PGD with $\epsilon = \frac{8}{255}$. Similarly, we approximate $\max_{||\delta||_2 \leq \sqrt{d} \epsilon} \ell(x+\delta)$ using  $\ell(x+\delta_2)$ where $\delta_2$ is generated using 100-iteration $\ell_2$-PGD with $\epsilon = 1.74$. The reason for this particular configuration of attack parameter is to match the ones used during our previous evaluations.
>
> Checkpoints             | $\ell(x+\delta_{\infty})$| $\ell(x+\delta_2)$
>
> Beginning of SOAR | 76.03          | 112.10
>
> End of SOAR             | 7.0              | 20.15
>
> From this preliminary study, we observe that there is indeed a gap between the approximated LHS and RHS of (7), and thus, we think it is a valuable future research direction to explore other possibilities that allow us to use a second-order approximation to study the worst-case loss subject to an $\ell_\infty$ constrained perturbation.
>
> Reference:
>
> Chris M Bishop. Training with noise is equivalent to Tikhonov regularization. Neural computation, 7(1):108–116, 1995

---

> ### Author Response · Authors · 2020-11-20
> **Response II**
>
> Response to comment 5:
>
> We included a comparison of the wall-clock time of different methods in Appendix E.6. In addition, we characterize the computation complexity as a function of the number of forward and backward passes required for a single mini-batch.
> Standard training: 1 forward pass and 1 backward pass
> Adversarial training with k-step PGD: k+1 forward passes and k+1 backward passes
> FOAR: 1 forward pass and 2 backward passes
> SOAR: 3 forward passes and 4 backward passes
>
> Response to comment 6:
>
> Consider a linear interpolation of the cross-entropy loss from $x$ to $x’$. Specifically, we consider $\ell(\alpha x + (1-\alpha)x’)$ for $\alpha \in [0,1]$. Previous work has empirically shown that the value of the loss behaves logistically as $\alpha$ increases from 0 to 1 [Madry et al. 2017]. In such a case, since there is very little curvature at $x$, if we use the hessian information exactly at $x$, it often leads to an inaccurate approximation of the value at $\ell(x’)$. Consequently, we have a poor approximation of the inner-max, and the derived regularization will not be effective.
>
> For the approximation in (12), this issue corresponds to the scenario in which the classifier is very confident about the clean input at $x$. In this case, both $r$ and $u$ will be close to zero. This means the approximated $\ell(x’)$ will not be accurate, and the regularizer becomes ineffective. We will clarify this idea in the revised version of the paper.
>
> Response to comment 7:
>
> We agree with the reviewer and will correct it in the revised paper. The error of the approximation is: $\frac{1}{2}\delta^\top (\nabla^2\ell(x+h^\star\delta) - \nabla^2\ell(x+h\delta))\delta$, meaning the quality of the approximation is related to the curvature information at x.
>
> Response to comments on the experiments 1:
>
> We notice that SOAR regularized models have shown greater vulnerabilities against AutoAttack. Since the standard AutoAttack consists of an ensemble of four different attacks, we tested our model against individual attacks and discovered that there is a considerable degradation in robust accuracy for targeted APGD-DLR and targeted FAB. This is an interesting observation and requires further investigation. We will modify the conclusion of the paper accordingly to reflect the new results.
>
> list of attacks:
>
> apgd-ce: untargeted APGD-CE
>
> apgd-dlr: targeted APGD-DLR
>
> fab: targeted FAB
>
> sa: square attack
>
> Results:
>
> Autoattacks(Linf norm with eps = 8/255) evaluated on CIFAR10:
>
> madry	apgd-ce:41.57%; apgd-dlr:38.99%; fab:39.68%; sa:47.87%
>
> trades	apgd-ce: 44.69%; apgd-dlr: 40.27%; fab: 40.64%; sa: 46.16%
>
> mart	apgd-ce: 45.01%; apgd-dlr: 39.22%; fab: 39.90%; sa: 46.90%
>
> mma	apgd-ce: 35.59%; apgd-dlr: 34.77%; fab: 35.50%; sa: 45.24%
>
> soar	apgd-ce: 53.40%; apgd-dlr: 18.25%; fab: 20.22%; sa: 35.94%
>
> foar	        apgd-ce: 31.15%; apgd-dlr: 27.56%; fab: 27.92%; sa: 35.92%
>
>
> Response to comments on the experiments 2:
>
> For baselines with WideResnet, we took all pretrained models from their public repositories. While evaluating adversarial training using PGD on Resnet10, we indeed experienced overfitting as the reviewer mentioned. Therefore, to ensure that the final model has the highest robustness, we save the model at the end of every epoch, and the final evaluation is based on the model with the highest PGD20 accuracy. This was discussed in Appendix E2.
>
> Reference:
>
> Aleksander Madry et al. Towards deep learning models resistant to adversarial attacks. arXiv preprint arXiv:1706.06083 (2017).
>
> *********************************
> Update:
>
> Autoattacks(L2 norm with eps = 0.5) evaluated on CIFAR10:
>
> standard	apgd-ce: 17.47%;	apgd-dlr: 1.98%; fab: 2.13%; sa: 41.94%
>
> madry	apgd-ce: 60.62%;	apgd-dlr: 58.78%; 	fab: 58.89%; sa: 65.81%
>
> trades	apgd-ce: 61.51%;	apgd-dlr: 58.78%; 	fab: 58.19%; sa: 68.17%
>
> soar	apgd-ce: 64.18%;	apgd-dlr: 52.97%; 	fab: 53.42%; sa: 71.29%
>
> *********************************
> Update: We have updated the paper with Section 5.3 to specifically discuss the evaluation with AutoAttack. We also modified other sections of the paper accordingly to reflect the new results.

---

> > ### Comment · AnonReviewer5 · 2020-11-23
> > **Response to the Authors**
> >
> > Thank you for the comprehensive response. I think many points are clarified, but I still think there are two main weakness of this paper.
> >
> > 1. *The tightness of the bound* The relaxation of (7) is too aggressive, which will render the resulting bound very loose. There should be a tighter bound for $\delta^T H \delta$ when $|\delta|_\infty \leq \epsilon$ by solving an integer programming. However, this naive method will be more computational expensive.
> >
> > 2. *Performance against the state-of-the-art* The experimental performance of SOAR is not competitive so far. The poorer performance of the model against APGD-DLR indicate gradient masking, especially the numeric issues caused by traditional cross-entropy loss. I encourage the authors to look into the loss landscape w.r.t the input, the curvature of PGD-trained model and SOAR-trained model should be different.
> >
> > Based on two points above, it is not convincing enough to argue the advantages of  SOAR over baselines like PGD adversarial training. This paper needs substantial edit and, from my point of view, not ready for publication in a premium conference like ICLR.

---

> > > ### Author Response · Authors · 2020-11-24
> > > **Response to the Reviewer**
> > >
> > > Thank you for your response.
> > >
> > >
> > > 1. Previous work shows that a mechanism based on gradient masking can be *completely* circumvented, resulting in a 0% or near-0% accuracy against non-gradient-based attacks (Athalye et al., 2018). Our result on SimBA and Square Attack shows that this is not the case with SOAR, even at $\epsilon = 8/255$. As such, the robustness improvement of SOAR cannot be *only* due to gradient masking.
> > >
> > >
> > > 2. Table 3 shows that, at $\epsilon = 4/255$, SOAR is 7~8% weaker than adversarial training on Targeted APGD-DLR and Targeted FAB outperforms, but it outperforms FOAR by a considerable margin against all attacks. This shows that SOAR is a reasonable extension to a first-order regularizer such as FOAR (Simon-Gabrial et al., 2019).
> > >
> > >
> > > 3. An idea or method does not need to be perfect in all aspects in order to play an important stepping stone in the scientific progress. SOAR is a well-motivated approach and is based on a well-justified second-order approximation. To our best knowledge, SOAR is currently the only regularization method (for both $\ell_\infty$ and $\ell_2$ attacks) motivated by a second-order approximation of the loss formulated under the robust optimization framework. The issue related to the looseness of the bound does not exist under an $\ell_2$ formulation of SOAR. The authors believe that the potential looseness of the bound for the $\ell_\infty$ formulation is an opportunity for designing better regularizers in the future, as opposed to an insurmountable problem justifying the rejection of this paper.
> > >
> > >
> > > 4. We have made substantial edits to the paper by addressing most of the concerns from the reviewers. In Section 5.3, we included a detailed discussion on the results with AutoAttack, as well as our hypotheses on the potential vulnerability of SOAR, which is important to be considered for future research.
> > >
> > > Reference:
> > >
> > > Athalye et al. Obfuscated gradients give a false sense of security: Circumventing defenses to adversarial examples. In International Conference on Machine Learning, 2018
> > >
> > > Simon-Gabrial et al. First-order adversarial vulnerability of neural networks and input dimension.  In International Conference on Machine Learning, 2019

---

> > > > ### Comment · AnonReviewer5 · 2020-11-24
> > > > **Response to the Authors**
> > > >
> > > > I thank the authors for the quick response. I agree with the authors that the second order defense method is an interesting direction to explore, and it indeed provides some sort of robustness. I need to clarify my claim that SOAR may result in **some sort of** gradient masking that makes its performance degrade in attacks like APGD-DLR/FAB attack. Gradient masking does not necessarily mean the defense method is completely broken. However, if a significant amount of the test instances suffer from gradient masking, we need to reconsider the training method.
> > > >
> > > > Based on the argument above, I do not think the contribution of proposing SOAR is enough, especially when considering the significant performance degradation under APGD-DLR/FAB (**10%+ gap below the baselines**). I do agree with the authors that a method does not need to be perfect under all cases, but the method here does really need to get polished when it is significantly worse in **standard benchmarks**.
> > > >
> > > > In small adversarial budget cases, SOAR is better than its first-order counterpart, but it is still worse than the classic adversarial training (ADV). In addition, *I think the performance of ADV is downplayed in Table 3*. According to https://arxiv.org/abs/2002.11569, if we use a validation set to do early stopping, the robust accuracy of a small ResNet model against PGD $l_\infty$ attack when $\epsilon = 8/255$ should be around 50%, **41.57% is too low to be convincing**. I encourage authors to check the experimental settings and results in https://arxiv.org/pdf/1706.06083.pdf and the leaderboard on https://robustbench.github.io/
> > > >
> > > > I agree with the authors that the bound in (7) is tighter when we consider $l_2$ adversarial budget, but this would be a limitation of this method, as $l_\infty$ robustness is more popular and better studied in this community.
> > > >
> > > > I strongly encourage authors to investigate how to derive a tighter bound of the second-order approximation and to overcome the problem of gradient masking. Should this problem be solved, it will be a nice paper.

---

### Author Response · Authors · 2020-11-23
**Summary of revision**

Dear reviewers,

Thank you very much for your careful reading of the paper and your valuable feedback! We have made our best effort to address your concerns, and we believe that the manuscript has improved a lot thanks to your feedback.

In the updated manuscript we have made the following changes:

- added a discussion on the connection between regularization and adversarial training, and emphasize that they are simply two different realizations of the same inner-max objective (End of Section 3)

- added an empirical study on how the number of sample sizes affects the regularized loss. (Appendix E.4)

- added a discussion on how an inaccurate Hessian estimation at x can result in an ineffective regularizer (Section 4.1)

- shortened the discussion on PGD white-box attacks to reflect the results using Autoattack(Section 5.1)

- added a discussion on results on Autoattacks: SOAR vulnerability, different epsilons, our hypotheses (Section 5.3, Table 3, Table 10)

- updated the discussion on the tightness of the bound in (7) (Appendix B.2)

- modified Conclusion to reflect the new results on AutoAttack.

- added a discussion on computation complexity (Appendix E.7)

All changes are in blue fonts to help better navigate and focus on the revised part of the paper.


During the ICLR rebuttal phase, we discovered that SOAR is vulnerable to AutoAttack (Croce et al., 2020), which consists of an ensemble of four different adversaries. We revised the paper to mention the performance of SOAR facing AutoAttack (Section 5.3). In addition, we have made substantial edits to the paper by modifying other parts of the paper to reflect the new results and addressing most of the concerns from the reviewers.

Through an extensive empirical evaluation with the four attacks in AutoAttack, we have identified the particular attack to which SOAR is vulnerable and provided hypotheses on SOAR's vulnerability. More importantly, we showed the robustness improvement of SOAR becomes more consistent among the four attacks at a smaller $\epsilon$, with robust accuracy surpassing the first-order counterpart. There are two implications. First, this shows that SOAR is a reasonable extension to a first-order regularizer such as FOAR (Simon-Gabrial et al., 2019). Second, combining the discussion with Reviewer 4 on the looseness of the bound in (7), the result shows that SOAR simply improves robustness against $\ell_\infty$ attacks with $\epsilon$ smaller than what it was formulated with.

Finally, the authors believe that this should not rule SOAR as a failed defense. Previous work shows that a mechanism based on gradient masking can be *completely* circumvented, resulting in a 0% or near-0% accuracy against non-gradient-based attacks (Athalye et al., 2018). Our result on SimBA and Square Attack shows that this is not the case with SOAR, even at $\epsilon = 8/255$. As such, the robustness improvement of SOAR cannot be *only* due to gradient masking.

An idea or method does not need to be perfect in all aspects in order to play an important stepping stone in scientific progress. SOAR is a well-motivated approach and is based on a well-justified second-order approximation. To our best knowledge, SOAR is currently the only regularization method (for both $\ell_\infty$ and $\ell_2$ attacks) motivated by a second-order approximation of the loss formulated under the robust optimization framework. The issue related to the looseness of the bound does not exist under an $\ell_2$ formulation of SOAR. The authors believe that the potential looseness of the bound for the $\ell_\infty$ formulation is an opportunity for designing better regularizers in the future, as opposed to an insurmountable problem justifying the rejection of this paper.

Reference:

Croce et al. Reliable evaluation of adversarial robustness with an ensemble of diverse parameter-free attacks. In International Conference on Machine Learning, 2020

Athalye et al. Obfuscated gradients give a false sense of security: Circumventing defenses to adversarial examples. In International Conference on Machine Learning, 2018

Simon-Gabrial et al. First-order adversarial vulnerability of neural networks and input dimension.  In International Conference on Machine Learning, 2019

---

### Decision · Program_Chairs · 2021-01-07
**Final Decision**

**Decision:**

Reject

**Comment:**

This paper proposes a regularization approach based on the second-order Taylor expansion of the loss objective to improve robustness of the trained models against \ell_inf and \ell_2 attacks. It is interesting to explore the second order-based regularization approach for network robustness. However, as pointed out by the reviewer, a major drawback of this approach is that SOAR is broken under a stronger attack - AutoPGD-DLR.  In addition, the theoretical bound seems very loose in the \ell_inf case.